# Lineage-specific differences and regulatory networks governing human chondrocyte development

**Daniel Richard**[1†], **Steven Pregizer**[2,3†], **Divya Venkatasubramanian**[2,3,4], **Rosanne M Raftery**[2,3], **Pushpanathan Muthuirulan**[1], **Zun Liu**[1], **Terence D Capellini**[1,5], **April M Craft**[2,3,6]*

[1]Human Evolutionary Biology, Harvard University, Cambridge, United States; [2]Department of Orthopedic Research, Boston Children's Hospital, Boston, United States; [3]Department of Orthopedic Surgery, Harvard Medical School, Boston, United States; [4]Department of Molecular and Cellular Biology, Harvard University, Cambridge, United States; [5]Broad Institute of MIT and Harvard, Cambridge, United States; [6]Harvard Stem Cell Institute, Cambridge, United States

*For correspondence:
april.craft@childrens.harvard.edu

†These authors contributed equally to this work

Competing interest: The authors declare that no competing interests exist.

**Abstract** To address large gaps in our understanding of the molecular regulation of articular and growth plate cartilage development in humans, we used our directed differentiation approach to generate these distinct cartilage tissues from human embryonic stem cells. The resulting transcriptomic profiles of hESC-derived articular and growth plate chondrocytes were similar to fetal epiphyseal and growth plate chondrocytes, with respect to genes both known and previously unknown to cartilage biology. With the goal to characterize the regulatory landscapes accompanying these respective transcriptomes, we mapped chromatin accessibility in hESC-derived chondrocyte lineages, and mouse embryonic chondrocytes, using ATAC-sequencing. Integration of the expression dataset with the differentially accessible genomic regions revealed lineage-specific gene regulatory networks. We validated functional interactions of two transcription factors (TFs) (RUNX2 in growth plate chondrocytes and RELA in articular chondrocytes) with their predicted genomic targets. The maps we provide thus represent a framework for probing regulatory interactions governing chondrocyte differentiation. This work constitutes a substantial step towards comprehensive and comparative molecular characterizations of distinct chondrogenic lineages and sheds new light on human cartilage development and biology.

## Editor's evaluation

In this study the authors mapped chromatin accessibility in hESC derived chondrocyte lineages and mouse embryonic chondrocytes using ATAC-sequencing and revealed lineage-specific gene regulatory networks. They further validated the functional interactions of two transcription factors, Runx2 and RELA, with their predicted genomic targets. The significance of study is to help us understand chondrocyte differentiation mechanism.

## Introduction

Cartilage is a crucial component of the musculoskeletal system, providing structure and functioning in various capacities to support the pain-free movement. Chondrocytes are the cells that produce and maintain the collagen- and proteoglycan-rich extracellular matrix (ECM) of cartilage tissues throughout the body. In the appendicular skeleton, chondrocytes give rise to growth plate cartilage, a transient

tissue that provides a template for the elongation of endochondral bones, and articular cartilage, a permanent tissue that covers joint surfaces to allow for frictionless joint movement. Articular cartilage arises from the epiphyseal cartilage at the ends of the developing bones. Abnormal development of growth plate or articular chondrocytes can result in chondro- or skeletal dysplasias, while injury to and aging of chondrocytes can contribute to the development of joint degeneration (i.e. osteoarthritis). Pharmaceutical or gene-based treatments for the majority of these skeletal ailments are inadequate or simply do not exist due to the vast gaps in our knowledge regarding molecular mechanisms that govern the differentiation of chondrocytes into these two distinct lineages, especially in humans.

Much pioneering work focused on understanding the molecular regulation of chondrogenesis was performed in vivo using the mouse as a model system. A typical framework for these experiments involved genetic manipulation of a gene, either in the germline or conditionally, followed by careful phenotyping and gene expression readouts (e.g. in situ hybridization, immunohistochemistry, or quantitative polymerase chain reaction) to assess the effect on chondrocyte development. This paradigm led to the identification of signaling pathways such as TGFB/BMP (reviewed here *Wang et al., 2014*), IHH (*Long et al., 2001*; *St-Jacques et al., 1999*), and PTHrP (*Karaplis et al., 1994*; *Lanske et al., 1996*), as well as TFs such as Sox5/6/9 (*Bi et al., 1999*; *Akiyama et al., 2002*), Runx2/3 (*Inada et al., 1999*; *Kim et al., 1999*; *Yoshida et al., 2004*), MEF2C (*Arnold et al., 2007*), HIF2a (*Schipani et al., 2001*), and FOXA2/3 (*Ionescu et al., 2012*) that are important for development growth plate and/or articular chondrocytes. Mouse models have indeed substantially contributed to our evolving understanding of gene functions and related diseases, as human mutations often recapitulate murine phenotypes and vice versa.

A growing number of studies within the past several decades have attempted to build on these seminal findings by exploring gene expression and gene regulatory mechanisms in chondrocytes on a broader scale. A significant number of these studies have focused on the Sox family, especially Sox9 (*Ohba et al., 2015*; *He et al., 2016*; *Liu and Lefebvre, 2015*; *Oh et al., 2014*; *Oh et al., 2010*). Early studies employed then-emerging technologies such as ChIP-chip (*Oh et al., 2010*) and expression microarrays (*Lui et al., 2015*; *Chau et al., 2014*; *Yamane et al., 2007*; *Tan et al., 2018*; *James et al., 2005*; *Cameron et al., 2009*; *James et al., 2010*), while later studies incorporated ChIP-seq and/or RNA-seq (*Ohba et al., 2015*; *He et al., 2016*; *Liu and Lefebvre, 2015*; *Oh et al., 2014*; *Duan et al., 2020*; *Li et al., 2017*; *Vail et al., 2020*; *Cheung et al., 2020*; *Zhang et al., 2021*) as well as ATAC-seq (*Guo et al., 2017*). Chondrocytes used for these experiments were derived from a variety of sources including rib or epiphyseal cartilage from embryonic, neonatal, or juvenile rodents (*Lui et al., 2015*; *Chau et al., 2014*; *Yamane et al., 2007*; *Tan et al., 2018*; *James et al., 2005*; *Cameron et al., 2009*; *James et al., 2010*; *Duan et al., 2020*; *Guo et al., 2017*), as well as a rat chondrosarcoma cell line (*Liu and Lefebvre, 2015*; *Oh et al., 2010*). These cells all serve as generally good models of growth plate development, but not necessarily articular chondrocyte development. Modeling articular chondrocyte development using rodent-derived cells is inherently challenging due to the limited availability of the source tissue and the failure of the cells to retain their phenotype during expansion in culture. Chondrocytes from neonatal bovine articular cartilage are more plentiful and have been used in at least one gene-regulatory study (*Zhang et al., 2021*). A handful of studies have used chondrocytes derived from human sources, including fetal epiphyseal cartilage (*Li et al., 2017*; *Vail et al., 2020*) and mesenchymal stromal cell-derived cartilage (*Vail et al., 2020*; *Cheung et al., 2020*). While helpful for illuminating nuances of human growth plate chondrocyte development, and albeit understandably less so for articular chondrogenesis, it is clear that we still lack a complete understanding of how distinct chondrogenic lineages are specified and maintained in humans.

Using a directed differentiation approach inspired by embryonic chondrogenesis, we differentiate human pluripotent stem cells (i.e. hESCs/iPSCs) into growth plates and articular chondrocytes (*Craft et al., 2015*). Following the induction of appropriate mesoderm and mesenchymal-like progenitors, chondrogenesis is induced in a high-density micromass format, eventually producing disks of cartilage tissues approximately 1 cm in diameter and 1–3 mm thick. Long-term culture of the micromasses with TGFβ3 results in the generation of articular-like cartilage tissue, while a transition to long-term treatment with BMP4 results in the growth of plate-like cartilage tissue. We previously defined 12 weeks as the end-stage of this micromass protocol, where the cells and tissues exhibit key characteristics of their in vivo counterparts, including the morphology and size of the cells, tissue/zonal organization, proteoglycan content, and expression of candidate cell-type specific markers such as lubricin

(encoded by *PRG4*) and type X collagen (encoded by *COL10A1*). Lubricin, produced by superficial zone chondrocytes in articular cartilage, provides boundary lubrication and reduces friction between articulating cartilage surfaces (*Lee et al., 2018*). Type X collagen is specifically expressed by hypertrophic chondrocytes in growth plate cartilage. This directed differentiation platform thus provides an opportunity to investigate human chondrogenesis and cartilage development in vitro.

We used our established in vitro hPSC-based model of developing human cartilage to address the critical gaps in our knowledge of human cartilage development, with an important goal of providing a comprehensive guide of gene regulatory mechanisms governing articular versus growth plate chondrocyte cell fate. In this present study, we performed bulk RNA-sequencing (RNA-seq) in hESC-derived articular and growth plate cartilage to identify lineage-specific gene expression, and compared them to developing fetal cartilage. We also performed ATAC-sequencing (ATAC-seq) to define the accompanying regulatory landscapes in hESC-derived chondrogenic lineages and in mouse chondrocytes isolated by cell sorting (*Guo et al., 2017*; *Richard et al., 2020*). Integrating the transcriptomic and epigenetic datasets suggests gene regulatory networks specific to the growth plate or articular chondrocyte lineages. For two such networks, RUNX2 and RELA, we provide evidence of transcription factor interaction with predicted target gene regulatory elements, validating the predictive gene regulatory networks uncovered by our analyses of these two distinct human chondrogenic lineages.

## Results

### Transcriptomic profiles of hESC-derived chondrocytes recapitulate those of human fetal chondrocytes

Having differentiated hESCs to produce articular chondrocytes and growth cartilage chondrocytes (*Figure 1A*), we performed bulk RNA-seq on these cartilage tissues after 12 weeks in culture. To serve as in vivo references for the in vitro cartilage tissues, we performed bulk RNA-seq on human distal femur articular and growth plate chondrocytes isolated from embryonic day (E)67 fetal donor tissue (*Supplementary file 1a-b*). Principal component analysis indicated the four different chondrocyte sources (hESC-derived growth plate cartilage, hESC-derived articular cartilage, fetal growth plate cartilage, and fetal articular) clustered separately (*Figure 1B*). Reassuringly, even though hESC differentiation and RNA-seq were performed on more than one occasion (4 independent differentiations and 3 sequencing batches), transcriptome clustering was primarily dependent on cell type. That is, hESC-derived articular (orange icons) or growth plate tissues (light blue icons) from different experiments and batches cluster together, indicating reproducibility within the hESC model system. Principle component 1 (PC1) generally represents differences observed between the in vitro and in vivo samples (*Figure 1B*, circles/triangles vs. squares), including minor contributions from sex-linked genes (hESC-derived samples are female and the fetal donor tissue was male). The top genes contributing to PC1 are enriched in GO biological processes that, with the exceptions of extracellular matrix/structure organization and immune responses, are detecting cellular responses to ions (*Figure 1—figure supplement 1*), suggesting differences associated with culturing cells in media not identical to the milieu in vivo. PC2 represents differences between articular/epiphyseal and growth plate cartilages, which were more pronounced for hESC-derived chondrocytes than for their in vivo counterparts, as indicated by the greater distance in separation along the PC2 axis. The top genes contributing to PC2 are enriched in GO biological processes that are consistent with cartilage and skeletal system development and morphogenesis (*Figure 1—figure supplement 1*).

The top 40 differentially-expressed genes (DEGs) between respective cartilage lineages, in vitro or in vivo, are shown in *Figure 1C–D*, respectively (all DEGs are provided in *Supplementary file 1b*). We performed gene-set enrichment analyses on the set of genes upregulated in hESC-derived articular or hESC-derived growth plate cartilage. The former was enriched for terms relating to ECM organization, response to TGF stimulus, and collagen processes, while the latter was enriched for terms relating to ossification, ECM organization, and cartilage development (*Supplementary file 1c*). We obtained similar enrichment terms when we performed the same analysis on genes upregulated in fetal epiphyseal or fetal growth plate cartilage (*Supplementary file 1c*). Of the top 200 genes with the highest degree of differential expression between hESC-derived articular and growth plate chondrocytes, >70% exhibited similar differential expression in fetal epiphyseal and fetal growth plate chondrocytes, a trend that was statistically significant (*Figure 1E*, compare top and bottom graphs;

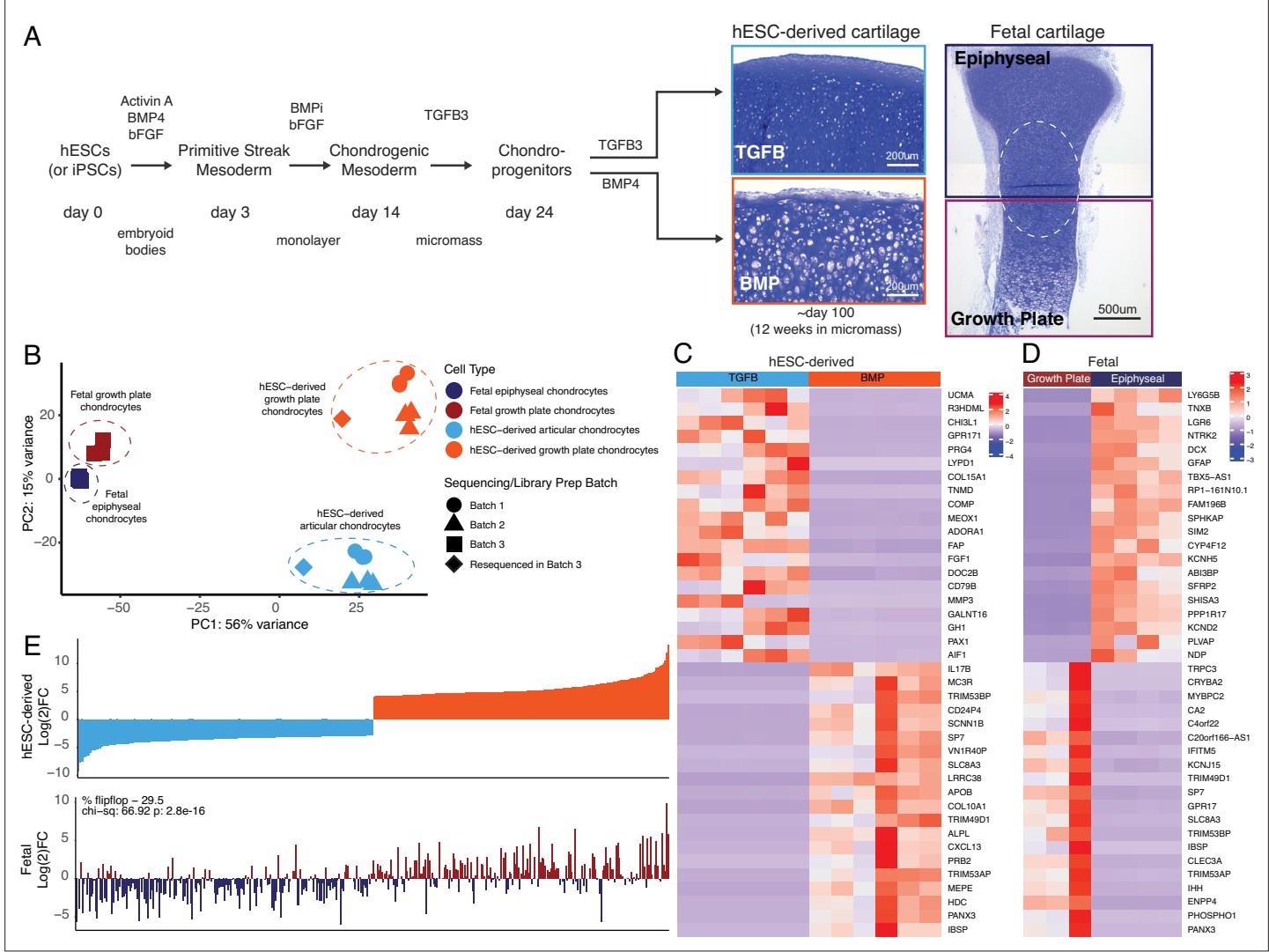

**Figure 1.** hESC-derived articular and growth plate chondrocytes have distinct transcriptional profiles that are similar to their respective fetal cartilage counterparts. (**A**) Brief methods to generate hESC-derived cartilage accompanied by toluidine-blue staining sections of hESC-derived articular (TGFB) and growth plate-like (BMP) cartilage tissues and the epiphyseal and growth plate cartilage of a developing fetal cartilage (E59 proximal tibia) show distinct chondrocyte morphology and proteoglycan-rich matrix. Fetal dissection location noted is approximate, and the dotted circle highlights the overlap of chondrocytes with similar phenotypes likely present in both samples following dissection. (**B**) PCA plot of RNA-seq expression data from hESC-derived and fetal cartilages. Legend indicates cell type and sequencing batch. (**C**) Expression heatmap of the top 20 differentially-expressed genes (DEGs) up- and down-regulated when comparing hESC-derived articular and growth plate cartilage. Red/blue color scale indicates Z-score expression values across samples in each plot. Columns indicate biological replicates. (**D**) Expression heatmap of the top 20 DEGs up- and down-regulated when comparing fetal epiphyseal and growth plate cartilage tissues. Red/blue color scale indicates Z-score expression values across samples in each plot. Columns indicate biological replicates. (**E**) The top 100 DEGs up- and down-regulated in the hESC-derived cartilages (top) were compared with equivalent log(2)FC values from the fetal cartilage (bottom).

The online version of this article includes the following figure supplement(s) for figure 1:

**Figure supplement 1.** Gene-set GO biological process enrichments for the top 200 gene loadings of principle component 1 (PC1) and PC2.

**Figure supplement 2.** Direction sharing of top differentially-expressed genes (DEGs) (by log(2)FC or p-value) between in vitro-derived and primary tissue samples.

**Figure supplement 3.** hESC-derived articular and growth plate chondrocytes have distinct transcriptional profiles at early, mid, and late stages of differentiation.

**Figure supplement 4.** Transcription factor expression and overlap between hESC-derived and human fetal chondrocytes.

p=2.8e−16, *Supplementary file 1d*). Notable genes whose expression in fetal cartilage is opposite that of hESC-derived cartilage are *MMP13*, a collagenase often observed in pathogenic osteoarthritic cartilage, and *SCARA5*, a dexamethasone-responsive gene implicated in adipogenesis (*Lee et al., 2017*). We obtained similar results in expression trends when starting with the top 200 genes with the highest degree of differential expression between fetal epiphyseal and fetal growth plate chondrocytes, and when considering those DEGs whose difference is most significant from either dataset (*Figure 1—figure supplement 2*). Taken together, these data lend strong support to the notion that the two chondrogenic cell types derived from hESCs represent *bona fide* articular and growth plate chondrocytes.

## Differentially expressed transcripts localize to specific regions within hESC-derived and fetal articular and growth plate cartilages

RNA-seq differences between fetal and hESC-derived cartilages may represent differences in the relative abundance of specific chondrocyte subtypes between the two samples. For example, mature articular cartilage has a superficial, intermediate, deep zone, and calcified chondrocytes, but fetal epiphyseal cartilage is less mature and contains surface chondrocytes that will give rise to the future articular cartilage as well as chondrocytes that will contribute to the secondary ossification center (future growth plate chondrocyte) (*Lui et al., 2015*; *Chau et al., 2014*). Growth plate cartilage has resting, proliferating, pre-hypertrophic, and hypertrophic chondrocytes. But as the developing cartilage is a continuous unit in vivo, the fetal epiphyseal chondrocytes and fetal growth plate both contain a portion of resting and proliferating chondrocytes, indicated by the approximate dissection point and dashed circle in *Figure 1A*. Therefore, we used in situ hybridization to localize sites of differential gene expression in hESC-derived and fetal cartilages and to estimate the fraction of cells expressing those transcripts (*Figure 2*). Type II collagen, encoded by the gene *COL2A1*, is a major structural component of both articular and growth plate cartilage, and as such, expression is observed in the cartilaginous structures both in vitro and in vivo (*Figure 2A–D*). *PRG4* is expressed in the superficial layer of the hESC-derived TGFB-treated articular cartilage (*Figure 2A*) and absent in the BMP4-treated growth plate cartilage (*Figure 2B*). Similarly, in vivo, *PRG4* is expressed in the superficial zone of fetal articular cartilage, as well as the intra-articular ligaments and meniscus (*Figure 2C*), and is absent in the growth plate (*Figure 2D*). Tenomodulin (*TNMD*), a well-known marker of tendon fate (*Docheva et al., 2005*), was a top DEG in hESC and fetal articular chondrocytes, where it is found in cells at the most superficial layers of the hESC-derived articular cartilage and the fetal knee cartilage (*Figure 2A and C*, white arrows). *TNMD* is also expressed in the intra-articular ligaments in vivo, as expected (*Figure 2C* double arrow). *COL10A1* mRNA is detected in the hESC-derived growth plate cartilage (*Figure 2F*), but not hESC-derived articular cartilage (*Figure 2E*), consistent with expression patterns found in the fetal knee, where *COL10A1* is expressed in the hypertrophic chondrocytes of the growth plate (*Figure 2H*) but not in the epiphyseal chondrocytes (*Figure 2G*). Thus, lower fold-change differences in genes (e.g. *PRG4*) in the fetal cartilages compared to the hESC-derived cartilages may reflect the incomplete terminal differentiation in the fetal cartilage, while others (e.g. *COL10A*) may reflect higher proportions of hypertrophic chondrocytes among the cells recovered from hESC-derived growth plate cartilage compared to the fetal growth plate.

Additional DEGs were validated by qPCR in hESC-derived chondrocytes from five additional independent differentiations (*Figure 2I*), and fetal chondrocytes from the distal femur and proximal tibia of three developmental timepoints (E59 (Carnegie Stage 23), E67, and E72; *Figure 2J*). As predicted by previous studies (*Karaplis et al., 1994*; *Hagan et al., 2019*; *Nakajima et al., 2003*; *Ellman et al., 2013*; *Davidson et al., 2005*; *Martin, 2016*; *Iwamoto et al., 2010*; *Miao and Scutt, 2002*), expression levels of *FGF18* and *PTHLH* are significantly higher in hESC-derived articular chondrocytes, and levels of *FGFR3*, *PTH1R*, *PANX3*, and *ALPL* are significantly higher in the hESC-derived growth plate chondrocytes. Similar patterns are observed in fetal chondrocytes.

The transcriptomic data from hESC-derived cartilages identified genes that were not previously implicated in cartilage development in addition to confirming those that have been. We found *chitinase-3 like protein 1* (*CHI3L1*, also known as *YKL-40*) and mesenchyme homeobox 1 (*MEOX1*) to be top DEGs in the articular cartilage lineage and confirmed their lineage-restricted expression in vitro and in vivo (*Figure 1I–J*). *CHI3L1* expression has been described in cultured chondrocytes and osteoarthritic cartilage (*Knorr et al., 2003*; *Ling and Recklies, 2004*), however, *MEOX1* is most well-known

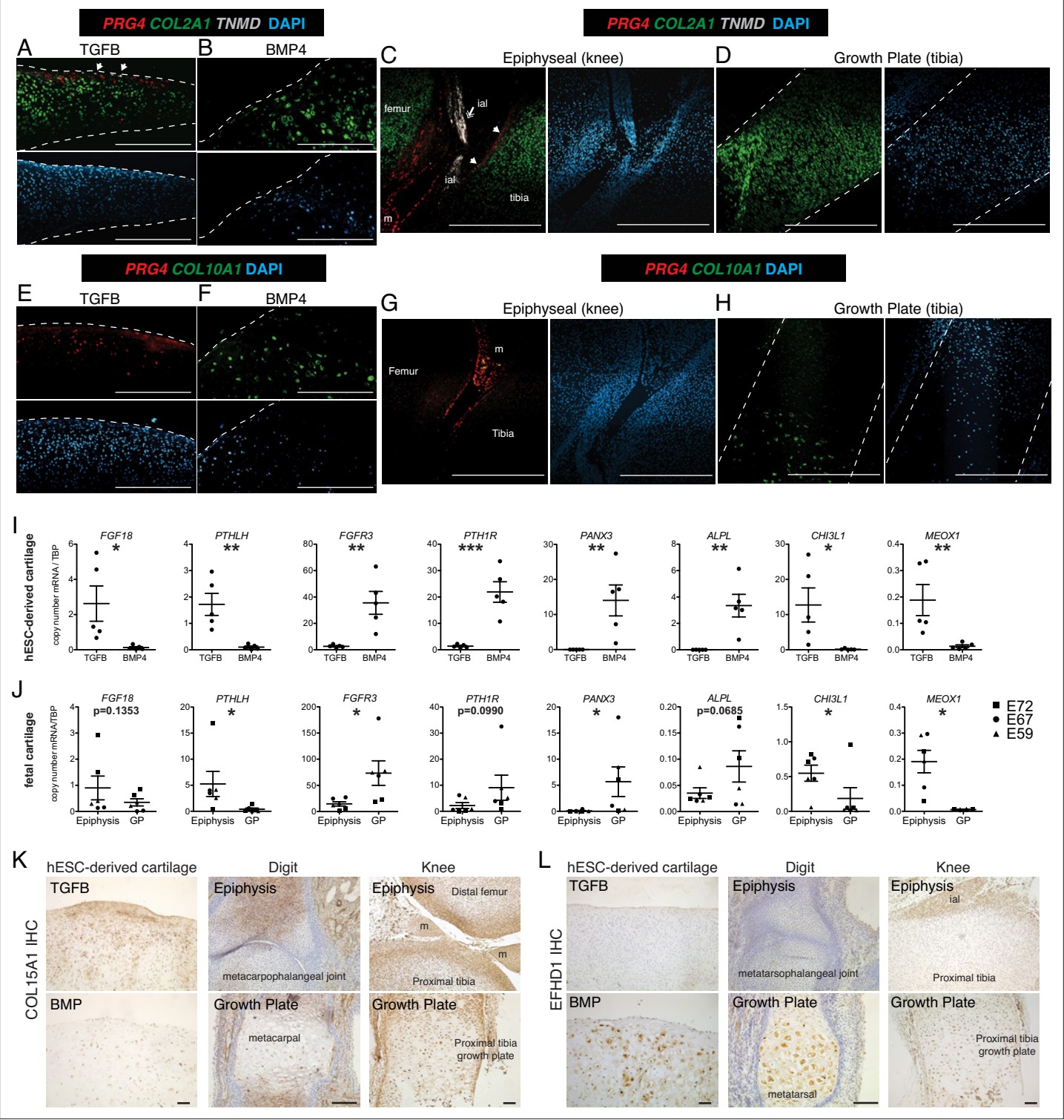

**Figure 2.** Validation of differential gene and protein expression in hESC-derived articular and growth plate cartilage and fetal epiphyseal and growth plate cartilage. (**A–H**) Confocal microscopy of hESC-derived and epiphyseal and growth plate fetal cartilage following in situ hybridization for indicated probes (RNAscope, *Wang et al., 2012*) and counterstained with DAPI (adjacent images). White arrows indicate *TMND* staining at the cartilage surface, and white double arrow indicates *TNMD* staining in an intra-articular ligament. Scale bar, 500 µm. ial, intra-articular ligament; m, meniscus (**I–J**) Quantitative RT-PCR of differentially expressed genes (DEGs) in hESC-derived cartilage (**I**, n=5 independent experiments with 3–6 replicates per experiment) and fetal cartilage (**J**). Chondrocytes were isolated from the epiphysis and growth plate (GP) of the distal femur and proximal tibia at E59 (triangles), E67 (circles), and E72 (squares). *p<0.05, **p<0.01, ***p<0.001, Student's t-test. Error bars, SEM. (**K–L**) Immunohistochemistry

*Figure 2 continued on next page*

*Figure 2 continued*

(IHC, brown staining) was used to validate the expression of indicated proteins within cartilage and joint tissues as indicated. Scale bar, 100 μm. ial, intra-articular ligament; m, meniscus. Sections counterstained with Mayer's hematoxylin (blue). Fetal IHC images are representative of at least three anatomical locations and two donor specimens; hESC-derived tissue IHC and in situ hybridization (**A–H**) are representative of tissues from at least three independent experiments.

for its role in somitogenesis and axial skeleton formation (*Skuntz et al., 2009*). *Type XV collagen* (*COL15A1*), differentially expressed in hESC-derived articular cartilage tissues but not fetal cartilages, is a non-fibrillar basement membrane-associated collagen previously detected in the perichondrium and in mesenchymal stem cells undergoing osteogenic differentiation (*Muona et al., 2002*; *Lisignoli et al., 2009*). The highest level of type XV collagen protein was localized in the matrix of the superficial zone of hESC-derived articular cartilage, and we detected intracellular staining in the deeper zone of the articular cartilage and in some cells within the growth plate cartilage (*Figure 2M*). We also localized Type XV collagen in developing human phalangeal (E70) and knee joints (E59), where we found it to be in the matrix of the epiphysis of the metacarpophalangeal joint, and at the surface of the knee joint cartilages, but absent in the matrix surrounding hypertrophic chondrocytes of the growth plates. These data suggest *COL15A1* expression may be specific to the superficial zone of articular cartilage. *EF-hand domain-containing protein 1* (*EFHD1*) expression was significantly higher in both hESC-derived and fetal growth plate chondrocytes. EFHD1 is a calcium-binding protein localized to the inner mitochondrial membrane, previously undescribed in cartilage (*Mun et al., 2020*). EFHD1 protein was localized to the cytoplasm of BMP4-treated hypertrophic chondrocytes, and hypertrophic chondrocytes in the fetal growth plates, but not in articular or epiphyseal cartilage, as predicted (*Figure 2N*). These data indicate *EFHD1* is specifically expressed in hypertrophic chondrocytes of the growth plate.

To validate the aforementioned gene expression patterns we identified in our analyses of 12-week-old cartilage tissues, and to explore differences between these lineages during their in vitro development over time, we performed bulk RNA-seq on three independent hESC-derived articular or growth plate cartilage tissues generated after 4, 8, or 12 weeks of TGFB or BMP4 treatment (henceforth referred to as the 'timecourse'). The top 40 DEGs at each timepoint are shown in *Figure 1—figure supplement 3* (all DEGs are provided in *Supplementary file 2a-c*). Despite batch differences associated with library preparation, time of sample acquisition, and smaller sample size, we found continuity between the transcriptomic profiles of the 12-week-old cartilage tissues in the timecourse and those acquired in our original analyses. We identified 1985 significant DEGs between the 12-week-old hESC-derived articular and growth plate tissues in the timecourse. Of those that were up-regulated in hESC-derived articular cartilage, 73.7% were also detected as significantly up-regulated in the same lineage in the original dataset. Similarly, 80.3% of the DEGs up-regulated in 12-week-old hESC-derived growth plate cartilage in the timecourse were also significantly up-regulated in the same lineage analyzed previously. When considering the top 200 most significant DEGs in the articular and growth plate cartilages in the timecourse, 91% and 96% of these were also significant in the corresponding lineage in the original dataset, respectively. As expected, the GO biological process associated with lineage-specific gene expression in the timecourse (*Supplementary file 2d and e*.g., extracellular matrix organization, cartilage development) is consistent with those enriched in the original tissues (*Supplementary file 1c*). These data collectively indicate a high level of reproducibility across many independent in vitro differentiations.

Gene expression differences were also observed between the developing articular and growth plate cartilage tissues after 4 weeks and 8 weeks of in vitro culture (*Figure 1—figure supplement 3* and *Supplementary file 1a-b*). A relatively smaller number of DEGs (841) were found to be significant after 4 weeks of differentiation towards the articular or growth plate lineage, 498 genes being more highly expressed in hESC-derived articular cartilage progenitors, and 383 being more highly expressed in hESC-derived growth plate cartilage progenitors (383). On the other hand, after 8 weeks of culture, 2268 DEGs were found to be significant, 1136 were up-regulated in the articular cartilage lineage and 1132 were up-regulated in the growth plate cartilage lineage. DEGs from both lineages at all timepoints (including the corresponding 12 weeks old tissues) were enriched in expected GO biological processes such as cartilage development/chondrocyte differentiation and skeletal system development (*Supplementary file 2d*). At 8 weeks, GO biological processes such as ossification and

endochondral bone morphogenesis were enriched in the hESC-derived growth plate cartilage, while hESC-derived articular cartilage became enriched in an extracellular matrix organization and response to TGFβ. As mentioned above, after 12 weeks, hESC-derived growth plate cartilage DEGs remained enriched in previous terms, and became enriched for replacement ossification, biomineralization, and bone morphogenesis, while the top processes in the articular cartilage lineage remained similar to earlier timepoints and the previous dataset.

The reproducible identification of known lineage-specific expression in the distinct cartilage tissues we generated from hESCs validates the utility of our established in vitro differentiation methods and further illustrates the strength of our transcriptomic datasets in identifying novel markers and potential regulators of articular and growth plate cartilage development.

## Chromatin accessibility differences between hESC-derived articular chondrocytes and hESC-derived growth plate chondrocytes

As cell fate decisions are guided by transcriptional regulation, we next sought to more deeply investigate the expression of TFs and potential gene regulatory elements within the hESC-derived cartilage tissues. From our initial differential gene expression analyses, we identified 277 TFs that were differentially expressed in at least one of the four cell types profiled and for which a binding motif has been described (*Supplementary file 1e*, *Figure 1—figure supplement 4*). Moreover, in our independent transcriptomic analysis of cartilage developing over time in vitro, we detected an additional 41 TFs that were differentially expressed in the 4 week cartilage tissues and 173 in the 8 week tissues, including those differentially expressed at more than one timepoint (*Supplementary file 2e-i* and *Figure 1—figure supplement 3*). To refine this list of potential chondrogenic lineage regulators, we performed ATAC-seq (*Buenrostro et al., 2015*), a method used to characterize chromatin accessibility on a genome-wide basis, on a subset of terminally differentiated hESC-derived chondrocytes that were used for transcriptomic analysis (i.e. the three biological replicate tissues per treatment sequenced in batch two, *Supplementary file 1a and f-g*). To establish a set of evolutionarily conserved and, therefore, likely functional regulatory elements in chondrogenesis, we also generated ATAC-seq data from mouse embryonic chondrocytes expressing *Col2a1* (expressed by all chondrocytes) or hypertrophic growth plate chondrocytes expressing *Col10a1*. Col2a1+or Col10a1+chondrocytes were isolated from E15.5 transgenic mice (i.e. stage-matched to our human embryonic samples) harboring fluorescent reporters driven by *Col2a1* or *Col10a1* regulatory elements using cell sorting (see Methods and previous work *Richard et al., 2020*). The genome-wide overlap of peaks found in the two types of human and mouse chondrocytes is summarized in *Table 1*. As hypertrophic growth plate chondrocytes can co-express both *Col2a1* and *Col10a1*, we expected there to be some overlap in peaks between the Col2a1+ sorted chondrocytes and the Col10a1+ chondrocytes. However, peaks identified in only Col10a1+ chondrocytes are expected to be more restricted to chondrocytes in the growth plate.

Profiling the hESC-derived chondrocytes by ATAC-seq and calling significant reproducible open-chromatin regions (i.e. peaks) revealed a total of 37,780 unique peaks, corresponding to putative regulatory elements. We categorized these regions on the basis of differential accessibility in either growth plate or articular chondrocytes, identifying 12,154 regions more accessible in growth plate chondrocytes and 11,571 more accessible in articular chondrocytes (*Supplementary file 3a-b*). These

**Table 1.** Summary of ATAC-seq peaks from mouse and human chondrocytes.

| | | | Mouse embryonic chondrocytes | | |
| --- | --- | --- | --- | --- | --- |
| | | | All | Col2+ | Col10+ |
| | | | **30,950** | **28,972** | **12,906** |
| | All Peaks (T+B) | 37,780 | 13,687 | | |
| | TGFB (All peaks) | 31,137 | | 13,381 | 9223 |
| | BMP (All peaks) | 29,821 | | 12,471 | 9070 |
| | TGFB (Unique) | 11,571 | | 3971 | 2385 |
| hESC-derived chondrocytes | BMP (Unique) | 12,154 | | 2584 | 1754 |

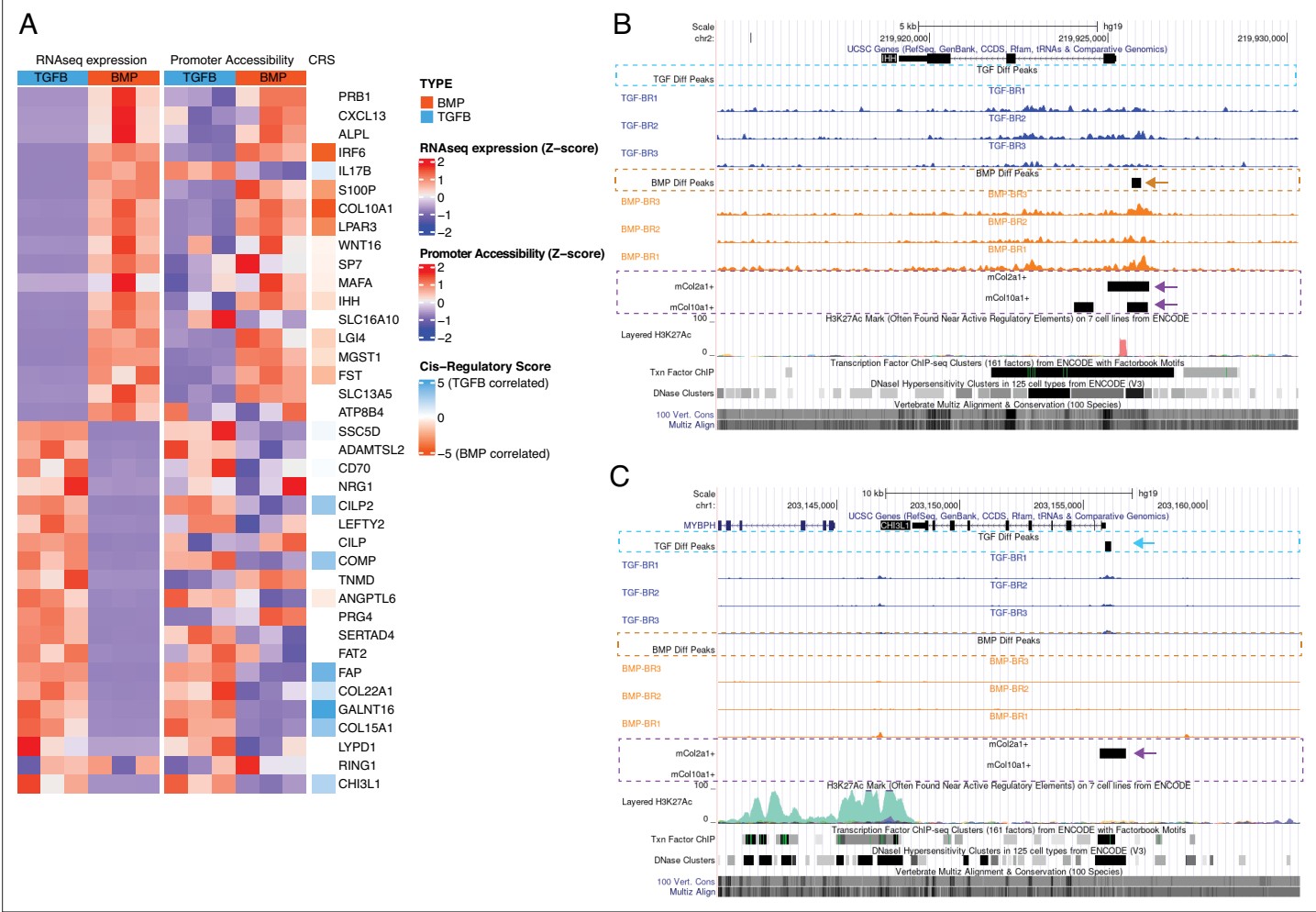

**Figure 3.** Epigenetic profiling of hESC-derived articular and growth plate chondrocytes. (**A**) The expression (left), gene-promoter accessibility (middle), and metric of cis-regulatory activity (cis-regulatory score, CRS, right) values of the top 20 differentially-expressed genes (DEGs) in each lineage. Red/blue color scale indicates Z-score expression/accessibility values across samples in each plot. Blue/orange scale indicates lineage-bias in cis-regulatory metric. (**B–C**) Representative differentially-expressed gene locus in each lineage show differentially accessible regions (DARs) at the promoter corresponding to the same lineage. IHH (**B**) is differentially expressed in the growth plate lineage while CHI3L1 (**C**) is differentially expressed in the articular cartilage lineage. Arrows highlight DAR of interest in respective tracks.

The online version of this article includes the following figure supplement(s) for figure 3:

**Figure supplement 1.** Top differentially expressed transcription factors (TFs) in hESC-derived ACs and GPCs.

**Figure supplement 2.** Motif enrichment is not correlated with sequence complexity.

differentially accessible regions (DARs) suggest cell-type specific regulatory activity and are the focus of subsequent analyses. To illustrate these data, the top 20 DEGs for each lineage in this subset of hESC-derived tissues, the accessibility of their corresponding promoters, and their respective *cis*-regulatory score (CRS, see below and *Methods*) are shown in **Figure 3A**. For example, DARs identified near the *IHH* and *CHI3L1* loci in hESC-derived growth plate chondrocytes and hESC-derived articular chondrocytes are indicated by black rectangles in the 'BMP Diff Peaks' track, orange dashed line and the 'TGF Diff Peaks' track, blue dashed line, respectively (**Figure 3B–C**). Tracks showing accessible regions detected in mouse embryonic chondrocytes are also shown (outlined with purple dashed lines) to visualize those regulatory elements that are conserved between species (examples indicated with arrows). Using the GREAT region-based association tool (**McLean et al., 2010**), we identified terms significantly associated with DARs from growth plate chondrocytes, including *anomaly of the limb diaphyses* and *ECM organization*. Likewise, we identified terms associated with DARs from

articular chondrocytes, including *ECM organization*, *collagen metabolic process,* and *osteoarthritis* (*Supplementary file 3c*).

## A simplistic model of gene expression is not sufficient to explain the gene regulatory network information captured by this merged transcriptomic and epigenetic data approach

To begin to understand differential transcriptional regulation mechanisms in these two lineages, we used de novo motif analysis to identify over-represented TF motifs in DARs specific to either articular or growth plate chondrocytes (*Supplementary file 3d-e*). We detected RNA expression for 15 TFs whose motifs are enriched in BMP DARs, but only five TFs were differentially expressed (*Supplementary file 1e and 3d*). *FOSL2* and *PITX2* were expressed at significantly higher levels in the corresponding BMP-treated growth plate lineage, while *SOX11*, *FOXA1*, and *RUNX1* were expressed at significantly higher levels in the opposite lineage. We detected RNA expression for 19 TFs whose motifs are enriched in TGFB DARs, but again only five of these were differentially expressed (*Supplementary file 1e and 3e*). *ETV4*, *AP4*, and *NFYB* were expressed at significantly higher levels in the corresponding TGFB-treated articular cartilage lineage, while *NFAT5* and *NHLH1* were expressed at significantly higher levels in the opposite lineage. Thus, the majority of motifs identified in these DARs were not for TFs that were also differentially expressed in the corresponding cell type.

We then examined the same two sets of DARs specifically for the enrichment of motifs belonging to TFs differentially expressed in the corresponding cell types. This yielded a reduced set of TFs, several of which were also observed in our de novo analysis (*Figure 3—figure supplement 1* and *Supplementary file 3f*). For this latter approach, we confirmed that motif enrichment is not substantially correlated with sequence complexity (*Figure 3—figure supplement 2*). When we considered sets of lineage-specific DARs nearby genes exhibiting lineage-specific expression, we observed similar enrichments for motif occurrences of several of these TFs in both region (lineage) sets, despite conditioning on the lineage-specific expression of these factors (*Figure 3—figure supplement 1*, right). For example, motifs for the top DE TFs, *POU2F2* (a TGFB-specific DE TF), and *MEF2C* (a BMP-specific DE TF) were significantly enriched in both TGFB-DARs and BMP-DARS when compared to randomly-sampled sequence sets (red lines, *Figure 3—figure supplement 1*, right). Motifs for only two of these DE TFs, *RUNX2,* and *RUNX3* were significantly enriched in the corresponding BMP lineage, and significantly depleted or not significant in the TGFB-DARs. This suggests that a simplistic model of gene expression, wherein upregulation of a given TF is associated with increased accessibility of elements to which it may bind, and subsequently increased expression of its putative targets, is not sufficient to explain the gene regulatory network information captured by our ATAC-seq/RNA-seq strategy.

## Defining hypothetical gene regulatory networks via cataloging gene expression and chromatin accessibility differences between hESC-derived articular chondrocytes and hESC-derived growth plate chondrocytes

We sought to integrate our ATAC- and RNA-seq datasets in a way that better captured the regulatory behavior described in our sequencing datasets. Our approach defined three metrics of expression and accessibility at a given locus: (1) gene expression, (2) proximal (promoter) accessibility, and (3) distal (enhancer) activity, defined as a *cis*-regulatory score (see Methods, *Supplementary file 4a*). Based on the simplistic model of gene expression described above, we would have expected absolute correspondence between these three metrics for all DEGs, however, there were clear deviations from this result (*Figures 3A and 4A*). We reasoned that multiple regulatory principles may be at play and, inspired by recent work describing the *cis*-regulatory behavior of immunological genes in mice (*Yoshida et al., 2019*), we classified genes into four different regulatory behaviors based on the proportion of variance in expression explained by chromatin accessibility within their respective loci. Briefly, these consist of genes whose expression variance is best explained by: variance does not clearly associate with chromatin accessibility ('unexplained,' cluster 1, *Supplementary file 4b*), a combination of promoter accessibility and distal *cis*-regulatory accessibility ('combo-centric, cluster 2, *Supplementary file 4c*), promoter accessibility alone ('promoter-centric,' cluster 3, *Supplementary*

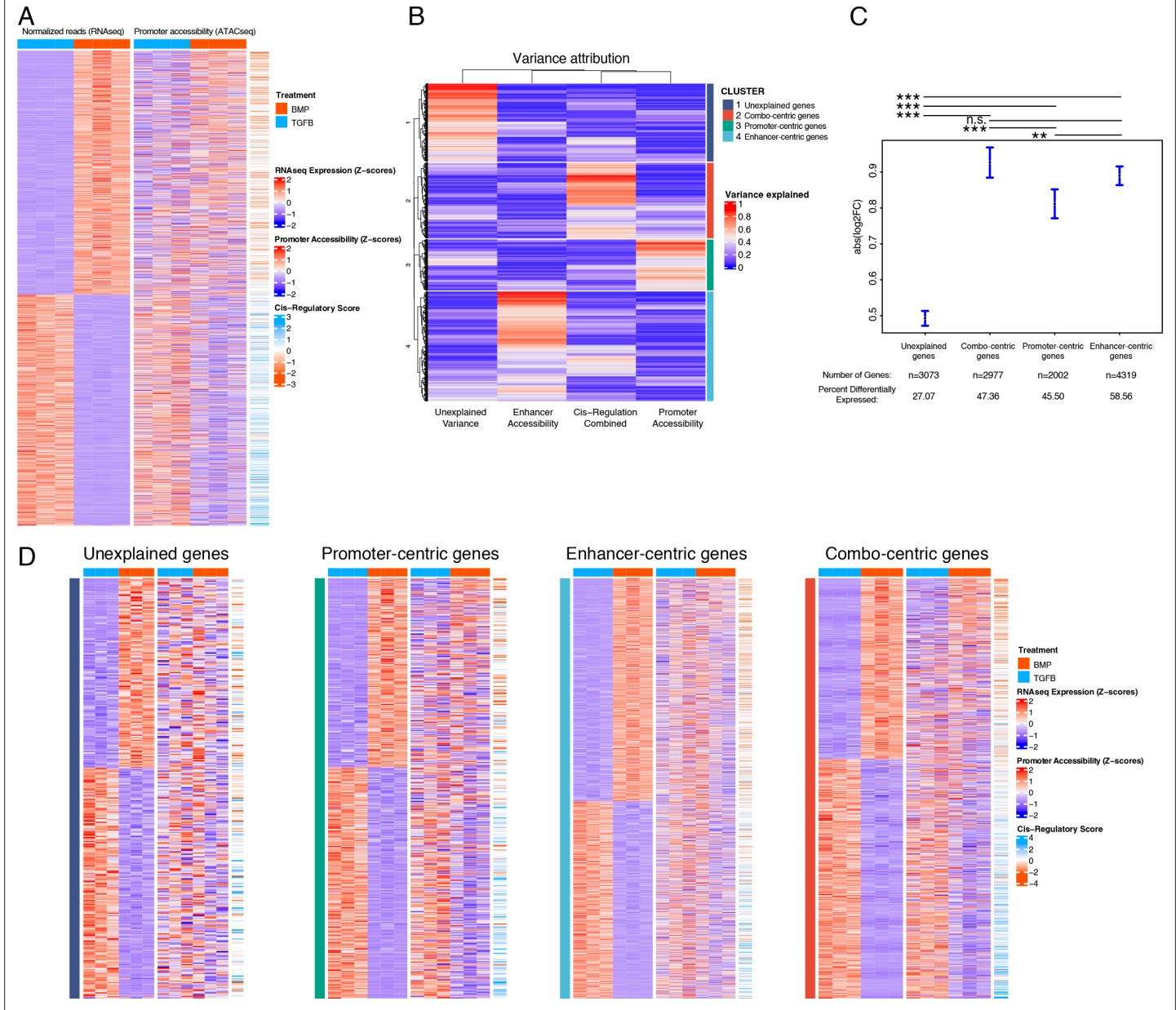

**Figure 4.** Variance in the expression of differentially-expressed genes (DEGs) can be attributed to different classes of regulatory elements (gene regulatory behavior). (**A**) Expression (left), gene-promoter accessibility (middle), and cis-regulatory metric (right) of all genes with expression logFC >1 across lineages. Red/blue, blue-orange color scale indicates Z-score of indicated metric across samples in each indicated plot. (**B**) For each gene in (**A**), the proportion of variance in expression which can be explained by regressing on individual accessibility metrics is shown in red/blue color scale (red = more variance). Hierarchical clustering dendrogram and cluster identity are shown on the left and right (respectively), indicating the four clusters of regulatory behavior identified. (**C**) LogFC values of genes clustered by regulatory behavior. Significance bars indicate Tukey post-hoc corrected p-values. Proportion of significant differentially expressed (DE) genes in each cluster is indicated (see *Supplementary file 4b-e*). n.s., not significant; *p<0.05; **p<0.01; ***p<0.001. (**D**) Similar plots to (**A**), for genes clustered by regulatory behavior. Within each heatmap genes are hierarchically clustered by expression logFC. Color scales as in (**A**).

*file 4d*), or distal *cis*-regulatory accessibility alone ('enhancer-centric,' cluster 4, *Supplementary file 4e*; *Figure 4B*). In general, genes falling into clusters 2–4 exhibited larger fold-changes in expression between articular and growth plate chondrocytes compared to genes falling into the 'unexplained variance' category (cluster 1; *Figure 4C*). Likewise, a greater proportion of genes from clusters 2–4 (genes whose variance in expression can be attributed to promoter or enhancer accessibility or both) were differentially expressed, compared to those from cluster 1 (whose variance cannot be attributed

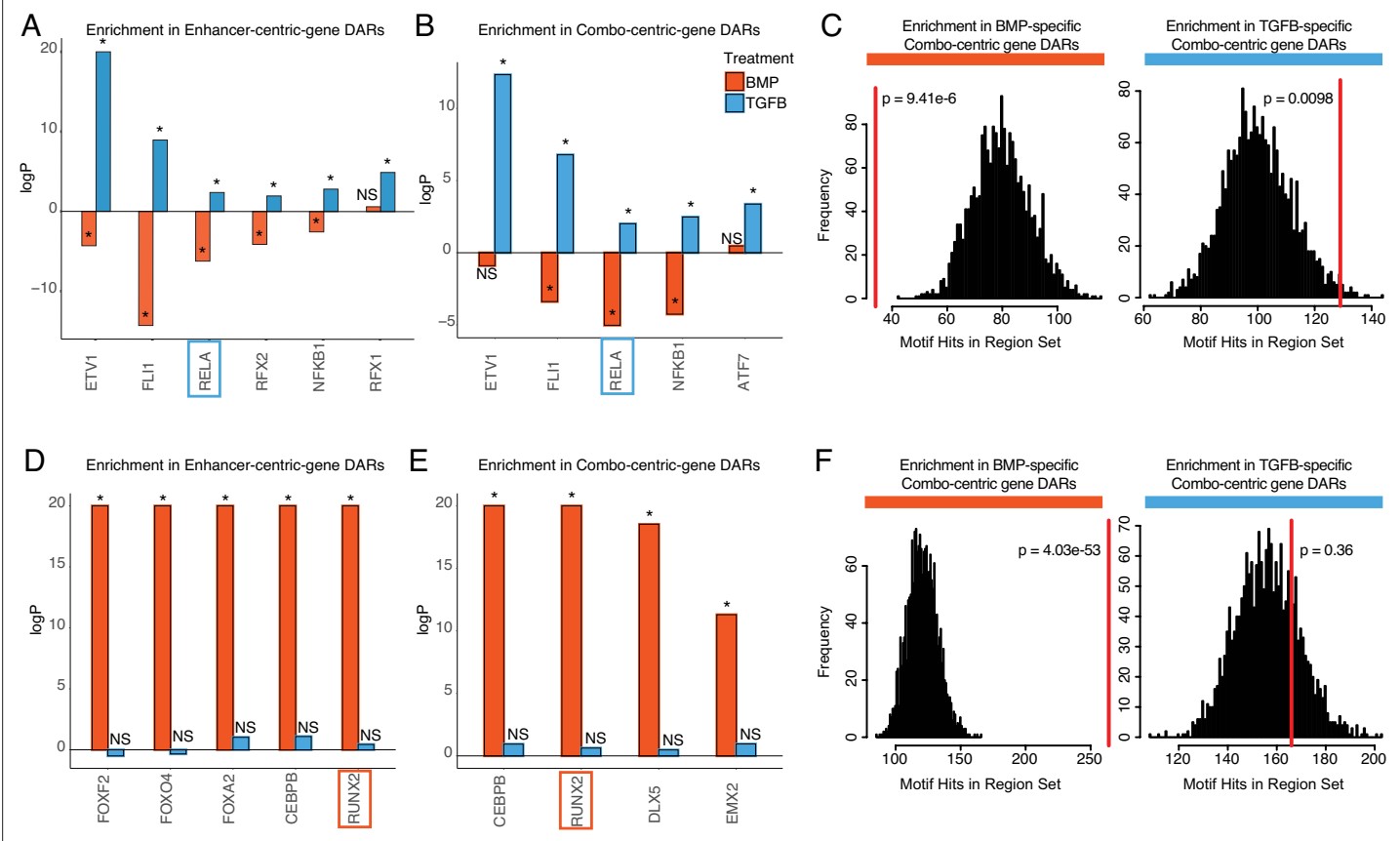

**Figure 5.** Identifying putative lineage-delineating transcription factors (TFs). Enrichment test results comparing the occurrence of the indicated motif in TGFB or BMP-biased differentially accessible regions (DARs) relative to randomized backgrounds. (**A**) TFs differentially expressed (DE) in TGFB-treated articular chondrocytes, testing motif occurrence in TGFB or BMP-biased DARs around enhancer-centric DEGs. *p<0.05; NS, not significant. (**B**) TFs DE in TGFB-treated articular chondrocytes, testing motif occurrence in TGFB or BMP-biased DARs around combo-centric differentially-expressed genes (DEGs). *p<0.05; NS, not significant. (**C**) Enrichment histogram of RELA motif occurrence in BMP (left) and TGFB (right)-biased DARs around combo-centric genes DE in their respective lineages. Red line indicates the target set value, black bars indicate occurrences in randomized sets. (**D**) TFs DE in BMP-treated growth plate chondrocytes, testing motif occurrence in TGFB or BMP-biased DARs around enhancer-centric DEGs. *p<0.05; NS, not significant. (**E**) TFs DE in BMP-treated growth plate chondrocytes, testing motif occurrence in TGFB or BMP-biased DARs around combo-centric DEGs. *p<0.05; NS, not significant. (**F**) Enrichment histogram of RUNX2 motif occurrence in BMP (left) and TGFB (right)-biased DARs around combo-centric genes DE in their respective lineages. Red line indicates the target set value, black bars indicate occurrences in randomized sets.

to differential accessibility in any putative regulatory elements, *Figure 4C*). Further, we confirmed that sets of genes segregated with this method show increased sharing of direction (i.e. lineage bias) for the expected parameters (e.g. 'combo-centric' gene expression had a greater correspondence with our *cis*-regulatory bias metric than did 'promoter-centric' gene expression) (*Figure 4D*).

We next looked for TF motif enrichment in the DARs of genes belonging to clusters 2–4 (i.e. combo-centric, promoter-centric, and enhancer-centric genes) with the following initial restrictions: (1) motifs were only considered for TFs that were differentially expressed between hESC-derived articular and growth plate chondrocytes (TGFB = 124, BMP = 83; *Supplementary file 1g and 3f*), and (2) enrichment of each motif was only considered for DARs, or promoters, in which the direction of accessibility (growth plate vs. articular) matched the direction of expression (growth plate vs. articular) (*Supplementary file 3f*). For each motif demonstrating enrichment according to these criteria, we then validated whether enrichment could not be significantly detected in the set of DARs/promoters for which the direction of accessibility was opposite to the direction of expression. This approach yielded a small number of motifs enriched in either promoter or enhancer sequences from cluster 2–4 genes (*Supplementary file 4f*), and that was biased towards articular chondrocytes or growth plate chondrocytes (*Supplementary file 4g-i*).

**Table 2.** Summary of candidate RELA targets.

| Putative Target | Chromosome location (hg19) | Distance from TSS of gene | Overlap with Mouse Col2a1+peaks | Overlap with mouse Col10a1+peaks | Validated ChIP-seq hits from other studies | Fold enrichment in ChIP-qPCR |
|---|---|---|---|---|---|---|
| PRG4 | chr1:186201240–186201490 | - 64.4 kb | Y | | 1 | 8.26 |
| LOXL2 | chr8:23268990–23269240 | - 7520 bp | | | 44 | 6.03 |
| LTBP2 | chr14:75083374–75083624 | - 4380 bp | | | 33 | 5.71 |
| GLIPR2 | chr9:36135932–36137932 | +10 bp | Y | | 17 | 5.57 |
| DKK3 | chr11:12101707–12101957 | - 71.1 kb | | | 134 | 11.56 |
| TLR2 | chr4:154577179–154577429 | +27.2 kb | Y | | 36 | 4.85 |
| COL15A1 | chr9:101733568–101733818 | +26.6 kb | Y | | 1 | 1.28 |

We focused on those motifs most highly enriched in DARs from enhancer-centric (cluster 4) and combo-centric (cluster 2) genes (*Figure 5*), as these groups exhibit the strongest trends in differential expression across lineages (*Figure 4C*). We identified seven TFs in the hESC-derived articular cartilage lineage whose expression and motif accessibility within enhancer-centric DEGs (*Figure 5A*) or combo-centric DEGs (*Figure 5B*) are significant and lineage-specific. *ETV1*, *FL1*, *RELA*, and *NFKB1* motifs are specifically enriched in both enhancer-centric and combo-centric gene DARs, while *RFX1* and *RFX2* motifs are enriched in enhancer-centric gene DARs, and *ATF7* motifs are enriched in combo-centric gene DARs. We used *RELA* as an example to illustrate these data. *RELA* motifs are significantly enriched in TGFB-specific combo-centric gene DARs (p=0.0098, *Figure 5C*), and significantly depleted in BMP-specific combo-centric gene DARs (p=9.41e-6). In the hESC-derived growth plate lineage, we identified seven TFs whose expression and motif accessibility within enhancer-centric DEGs (*Figure 5D*) or combo-centric DEGs (*Figure 5E*) are significant and lineage-specific. *CEBPB* and *RUNX2* motifs are specifically enriched in both enhancer-centric and combo-centric gene DARs, while *FOXF2*, *FOXO4,* and *FOXA2* motifs are enriched in enhancer-centric gene DARs, and *DLX5* and *EMX2* motifs are enriched in combo-centric gene DARs. Visualizing these results, *RUNX2* motifs are significantly enriched in BMP-specific combo-centric gene DARS (p=4.03e-53), but not significantly enriched in TGFB-specific combo-centric gene DARs (*Figure 5F*).

## Functional validation of TF and target interactions in human chondrocytes

To functionally validate the putative gene regulatory interactions we identified in these studies, we performed ChIP-qPCR for several enhancer and promoter elements assigned to DEGs that have putative binding sites for RELA or RUNX2 in hESC-derived chondrocytes (*Tables 2 and 3*). *RELA* and *NFKB1* are members of the same transcriptional complex and were also differentially expressed in the articular cartilage lineage at 8 weeks of differentiation (*Figure 1—figure supplement 3* and *Supplementary file 2b*). In the fetal donor samples, *RELA* and *NFKB1* were expressed at higher levels in the epiphysis compared to the growth plate, though the differences were not statistically significant. Having identified both of these genes in these conservative analyses, we postulated they have a cooperative functional role in articular chondrocyte biology and chose *RELA* as the differential p-value between the specificity of motif enrichments between hESC-derived articular cartilage DARs and growth plate DARs was higher than *NFKB1*. We also chose to investigate downstream *RUNX2* targets in the growth plate cartilage lineage, as it too was a differentially expressed TF in the growth plate lineage at 8 weeks, as well as in the fetal growth plate, and we wished to complement studies performed in the osteoblast lineages and in mice (*Wu et al., 2014*; *Hojo et al., 2021*) with human-specific data. We cross-referenced putative binding sites with ATAC-seq data collected from

**Table 3.** Summary of candidate RUNX2 targets.

| Putative Target | Chromosome location (hg19) | Distance from TSS of gene | Overlap with mouse Col2 +peaks | Overlap with mouse Col10 +peaks | Validated ChIP-seq hits from other studies | Fold enrichment in ChIP-qPCR |
|---|---|---|---|---|---|---|
| ACAN | chr15:89312870–89313120 | –33.0 kb | Y | | | 3.98 |
| ATOH8 | chr2:85969150–85969400 | –11.9 kb | Y | | 2 | 4.79 |
| C16orf72 | chr16:9166745–9166995 | –18.5 kb | | | 4 | 12.88 |
| COL10A1 | chr6:116439814–116440064 | +7110 bp | Y | Y | 4 | 2.44 |
| RCL1 | chr9:4837930–4838180 | +44.8 kb | | | 3 | 14.73 |
| WNT10B | chr12:49366141–49368141 | - 899 bp | | | 4 | 5.96 |
| GPR153 | chr1:6319685–6321685 | +699 bp | Y | Y | 3 | 4.06 |
| MAP4K3 | chr2:39719320–39719570 | –54.6 kb | Y | Y | 1 | 11.36 |
| RXRA | chr9:137178491–137178741 | –39.3 kb | | | 4 | 6.42 |
| SCUBE1 | chr22:43701886–43702136 | –36.8 kb | Y | Y | 1 | 9.27 |

E15.5 mouse Col2a1+ and Col10a1+ chondrocytes and published ChIP-seq data for several cell types (*Supplementary file 5a-e*). We chose 7–10 targets that satisfied some or all of these criteria, choosing some targets that have been previously described in chondrocyte biology, and others with binding sites that have overlapping ChIP-seq peaks in other cell types.

Seven putative RELA target loci (*Table 2*) were chosen to confirm by ChIP-qPCR, including several genes known to be involved in articular cartilage identity and maintenance. These include *PRG4* (lubricin), a functional marker for the superficial zone of articular cartilage; *LOXL2* (lysyl oxidase-like 2), which induces anabolic gene expression and plays a potential protective role against OA (*Alshenibr et al., 2017*); *DKK3* (Dickkopf-3), a noncanonical member of the Dkk family of Wnt antagonists that plays a role in articular cartilage maintenance (*Snelling et al., 2016*); and *TLR2* (Toll-like receptor 2), which mediates articular cartilage homeostasis (*Sillat et al., 2013*). We also chose to validate targets that are less well-studied, or newly identified in articular cartilage, including *LTBP2*, *COL15A1* (validated in *Figure 2M*), and *GLIPR2*. Representative binding regions with RELA motifs are the *GLIPR2* promoter (*Figure 6A*, overlapping with RELA ChIP-seq data and overlap with Col2a1+-mouse chondrocytes, indicated dotted box and red arrows) and an upstream enhancer of *LOXL2* (*Figure 6B*, overlapping with RELA ChIP-seq data and histone acetylation peaks). *RELA* and these putative target genes are expressed at significantly higher levels in hESC-derived articular cartilage (*Figure 6C–D*). The majority of RELA target genes were also expressed at significantly higher levels in the fetal epiphyseal chondrocytes compared to fetal growth plate chondrocytes (*Table 1* and data not shown), however, *RELA*, *COL15A1*, and *LOXL2* were not DEGs, likely due to under-represented terminally differentiated chondrocytes, and the overlap of unspecialized developing chondrocytes in both primary samples (*Figure 1A*, *Figure 2*). Notably, while *COL15A1* is not a DEG in the fetal chondrocytes, its protein expression appears higher in the matrix of the fetal epiphysis and superficial layers, compared to the matrix of the fetal growth plate (*Figure 2K*).

Following ChIP-mediated pulldown of genomic regions bound by RELA in TGFB-treated articular cartilage, four of the six loci were enriched at least fivefold (*PRG4, GLIPR2, DKK3, LOXL2, LTBP2*), and a fifth locus (TLR2) was enriched at least twofold compared to the negative control in the TGFB-treated articular cartilage (*Table 2* and *Figure 6—figure supplement 1*). The binding region for the

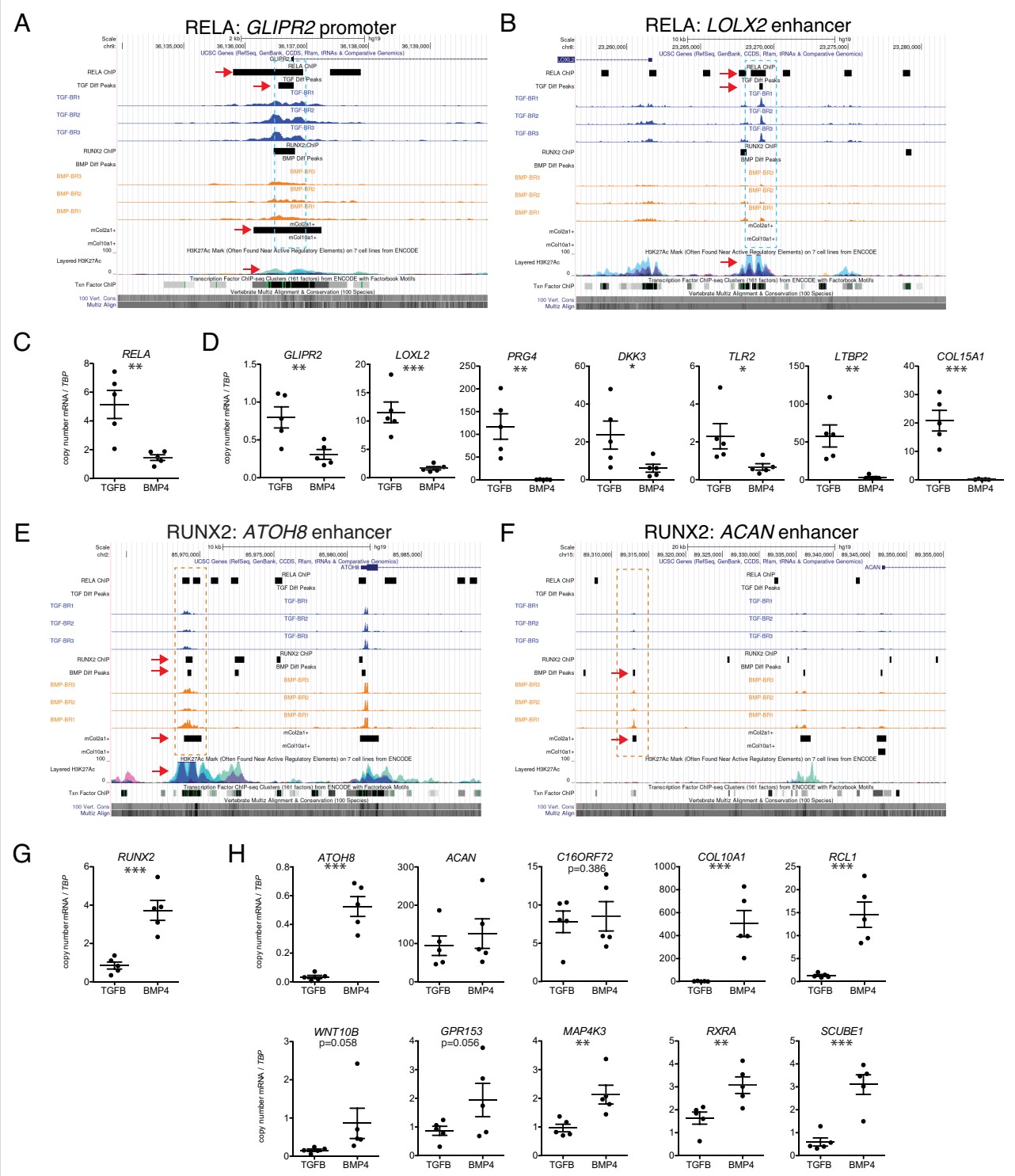

**Figure 6.** Putative targets of transcription factor (TF) regulation in hESC-derived articular and growth plate chondrocytes. (**A–B**) Two representative differentially accessible regions (DARs) (blue dashed boxes) in hESC-derived articular chondrocytes (TGFB-treated) which contain *RELA* binding motifs: near the promoter of *GLIPR2* (**A**) and an intronic enhancer of *LOXL2* (**B**). Region of interest indicated with the red arrow in relevant tracks. (**C**) *RELA* is differentially expressed in TGFB-treated articular chondrocytes, quantified by qRT-PCR. **p<0.01. (**D**) Expression of selected genes with putative *RELA* binding motifs was quantified by qRT-PCR. *p<0.05, **p<0.01, ***p<0.001, Student's t-test. Values indicate the mean of two to three biological replicates per five independent experiments. Error bars, SEM. (**E–F**) Two representative DARs (orange dashed boxes) in hESC-derived growth plate chondrocytes (BMP-treated) which contain *RUNX2* binding motifs: near the promoter of *ATOH8* (**E**) and an enhancer of *ACAN* (**F**). Region of interest indicated with the red arrow in relevant tracks. (**G**) *RUNX2* is differentially expressed in BMP-treatment, quantified by qRT-PCR. ***p<0.001. (**H**) Expression of selected

*Figure 6 continued on next page*

*Figure 6 continued*

genes with putative *RUNX2* binding motifs was quantified by qRT-PCR. *p<0.05, **p<0.01, ***p<0.001, Student's t-test. Values indicate the mean of two to three biological replicates per five independent experiments. Error bars, SEM.

The online version of this article includes the following figure supplement(s) for figure 6:

**Figure supplement 1.** Validation of transcription factor (TF) interaction with putative regulatory elements by Chromatin Immunoprecipitation – quantitative PCR (ChIP-qPCR).

*COL15A1* locus was only 1.3-fold enriched compared to the negative control in hESC-derived TGFB-treated articular cartilage, which suggests that RELA was not sufficiently bound to this locus in this sample.

Ten putative RUNX2 targets (*Table 3*) were chosen to confirm by ChIP-qPCR, including genes known to be important for chondrocyte and growth plate biology, including *ACAN* (Aggrecan), an essential proteoglycan in the extracellular matrix of both articular and growth plate cartilage (*Dateki, 2017*; *Lauing et al., 2014*) *COL10A1* (Type X collagen), a marker of hypertrophic chondrocytes important for endochondral bone formation (*Gu et al., 2014*); *WNT10B*, a Wnt family ligand thought to play a role in terminal chondrocyte differentiation and osteoblastogenesis (*Andrade et al., 2007*; *Bennett et al., 2005*); *ATOH8* (Atonal homolog 8), a transcription factor important for chondrocyte proliferation and differentiation in the cartilaginous elements of endochondral bone (*Schroeder et al., 2019*); and *RXRA* (Retinoid X receptor alpha), a retinoic acid receptor that plays a role in endochondral ossification (*Sun et al., 2019*). We also included targets previously undescribed in cartilage biology, including *C16ORF72*, *RCL1*, *GPR153*, *MAP4K3*, and *SCUBE1* based on previously described ChIP-seq interactions or homology with ATAC-seq peaks from mouse chondrocytes (*Table 3*). Representative gene regulatory elements with RUNX2 motifs are an upstream *ATOH* enhancer (*Figure 6E*, overlapping with RUNX2 ChIP-seq data, Col2a1+ mouse chondrocytes, and histone acetylation marks, indicated dotted box and red arrows) and an upstream enhancer of *ACAN* (*Figure 6F*), which overlaps with peaks found in mouse Col2a1+ chondrocytes and is homologous to an enhancer identified in mouse chondrocytes (*Li et al., 2018*). RUNX2 and the putative target DEGs are more highly expressed in hESC-derived growth plate cartilage (*Figure 6G–H*), with the exception of *ACAN* which is expressed in both cartilage lineages. Similarly, *RUNX2* and all but one putative target genes were more highly expressed in fetal growth plate chondrocytes compared to fetal epiphyseal chondrocytes, the exception being *C16ORF72* which was expressed at a similar level. Following ChIP-mediated pulldown of genomic regions bound by RUNX2 in BMP-treated growth plate cartilage, all 10 target loci chosen for validation were enriched at least twofold compared to the negative control (*Table 3* and *Figure 6—figure supplement 1*), confirming RUNX2 binding events at these gene regulatory elements. Six of the ten loci (*C16ORF72*, *RCL1*, *WNT10B*, *MAP4K3*, *RXRA*, *SCUBE1*) were enriched at least fivefold compared to the negative control.

As the great majority of putative DARs, we predicted as harboring motifs recognized by RELA and RUNX2 in hESC-derived articular and hESC-derived growth plate cartilage, respectively, were indeed enriched compared to the negative control loci, we consider the utility of these datasets to be extremely valuable for further exploration of the molecular mechanisms underlying human chondrocyte biology and cell fate decisions.

## Discussion

We provide here unbiased molecular characterizations of both the transcriptomic signatures and gene regulatory landscapes of hESC-derived articular and growth plate chondrocytes. We also provide evidence that these hESC-derived lineages are molecularly similar to their in vivo counterparts, through transcriptomic profiling of human fetal epiphyseal and growth plate chondrocytes, and epigenetic profiling of mouse embryonic chondrocytes that were isolated from either Col2a1-reporter (representing the majority of all mouse chondrocytes) or Col10a1-reporter mice (representing mouse growth plate chondrocytes). Specifically, we found strong correlations between hESC-derived articular cartilage and fetal epiphyseal samples, and likewise between hESC-derived growth plate cartilage and fetal growth plate samples.

We performed extensive experimental validation of DEGs, confirming lineage-specific patterns across multiple independent hESC differentiation experiments and primary cell datasets.

Receptor-ligand pairs, Fibroblast growth factor 18 (FGF18) and its receptor FGFR3 (*Hagan et al., 2019*; *Nakajima et al., 2003*; *Ellman et al., 2013*; *Davidson et al., 2005*), and Parathyroid hormone-like hormone (PTHLH) and its receptor PTH1R (*Karaplis et al., 1994*; *Martin, 2016*), are known to be differentially expressed between articular and growth plate cartilage, respectively. We found these expression patterns, and those of other known markers, to hold true in both the hESC-derived and fetal chondrocytes (*Figures 1–2*).

In addition to known markers, we identified novel genes that mark distinct cartilage lineages, such as *MEOX1* and *CHI3L1* in the articular cartilage lineage (*Figure 2I–J*), and *EFHD1* in the growth plate cartilage lineage (*Figure 2N*). *MEOX1* and *CHI3L1*, whose expression has been reported in the axial skeleton (*Martin, 2016*) and in osteoarthritic cartilage (*Knorr et al., 2003*), respectively, had not yet been identified in developing articular cartilage. EFHD1 was strongly localized to hypertrophic cells in hESC-derived and fetal growth plate chondrocytes. Previously studied in its role as a calcium sensor (*Hou et al., 2016*), EFHD1 could play a role in the mediating cellular response to calcium in hypertrophic chondrocytes (*Wang et al., 2001*). We also surprisingly found *tenomodulin* (*TNMD*) to be expressed in the superficial zone of articular cartilage (*Figure 2A and C*), co-expressed with *PRG4*. *TNMD*, closely related to *chondromodulin 1* (*CNMD*), is known as a functional marker for tenocytes (*Docheva et al., 2005*). While there is conflicting evidence of *TNMD* expression in resting and proliferating chondrocytes of the growth plate cartilage (*Shukunami et al., 2008*; *Brandau et al., 2001*), *TNMD* expression in the superficial zone of articular cartilage has not been previously described. Although this result was seemingly unexpected, both cartilage and tendons/ligaments rely on TGFB signaling, and they can arise from a common developmental progenitor (*Andrade et al., 2007*; *Pryce et al., 2009*; *Koyama et al., 2008*). This interesting finding warrants further exploration of the developmental relationship between cartilage and the adjacent connective tissues in the joint. Furthermore, these data uncover several other novel genes as yet unstudied in chondrocyte biology, underscoring the potential utility of tissue- or zone-specific markers, and the opportunity to investigate their function(s) in cartilage development or maintenance in human cells and other models.

Despite a strong overall transcriptomic and epigenetic correspondence between hESC-derived and primary cartilages, there are some limitations to our comparative analyses. One notable observation is the existence of genes whose expression patterns in hESC-derived cartilage lineages were opposite those seen in vivo fetal epiphyseal and growth plate cartilage tissues. This was an expected result, as the cartilage dissected from fetal samples is more heterogeneous than the hESC-derived tissues, contains fewer terminally differentiated specialized chondrocytes, and also likely has significant overlap in the composition of resting and proliferative chondrocytes. For example, the dissected epiphyseal cartilage includes perichondrium, resting zone chondrocytes, proliferative chondrocytes, in addition to chondrocytes that will participate in events related to the secondary ossification center and, perhaps in less abundance, those that will eventually give rise to the neonatal and adult articular cartilage. Likewise, the growth plate cartilage includes proliferative, pre-hypertrophic, and hypertrophic chondrocytes, in addition to perichondrium cells (our micro-dissection approach aimed to omit osteoblasts and hematopoietic cells). We also found differences when we compared the epigenetic profiles of mouse embryonic chondrocytes, expressing either *Col2a1* or *Col10a1* to those of hESC-derived chondrocytes, which to some extent were anticipated due to species specificity of genomic regulatory elements. However, Col2a1+ sorted chondrocytes encompass all types of chondrocytes, including both articular and growth plate chondrocytes. As such, the mouse epigenetic profiles reported herein do not accurately reflect a clear distinction between articular and growth plate cartilage lineages. The peaks we found to be conserved between the human and mouse chondrocytes, therefore, likely represent biologically relevant regulatory elements driving chondrogenesis. Finally, while our transcriptomic and epigenetic investigation of developing hESC-derived articular and growth plate cartilage effectively identified known regulators of and genomic regulatory elements important for chondrogenesis, additional mechanistic studies, such as the use of transgenic mouse models, are required to demonstrate function and necessity of novel targets.

It remains unclear where hESC-derived articular and growth plate cartilage lies in developmental time relative to fetal cartilage. Some obvious differences are the detection of latestage growth plate marker gene expression such as Integrin Binding Sialoprotein (IBSP) in eight week and 12-week-old hESC-derived growth plate chondrocytes, which was lacking in the fetal growth plate chondrocytes and the 4-week-old hESC-derived growth plate chondrocytes. We also demonstrated the presence

of a distinct superficial zone of cartilage in the hESC-derived articular cartilage, which is less developed and also less abundant in the fetal tissue used in this study (i.e. we indicated that the surface of the epiphysis corresponded to the site of the future superficial zone of articular cartilage). The potentially more developed/mature superficial zone in the hESC-derived articular cartilage may explain why superficial-zone-specific genes, such as *COL15A1*, a non-fibrillar basement membrane-associated collagen (*Clementz and Harris, 2013*), were identified as differentially expressed in the hESC-derived articular cartilage but not in the fetal epiphyseal cartilage, despite protein localization being lineage-specific (*Figure 2K*). Future studies focused on transcriptomic profiling of chondrocytes at the single-cell level, from in vitro-derived tissues and primary tissues, will address some of these standing questions.

Given our confidence in the divergent properties of the hESC-derived chondrogenic lineages, and that they reflect in vivo lineage properties, we also sought to use this system to define putative gene-regulatory networks (GRNs) which may govern lineage specification and gene expression patterns in developing chondrocytes. As TFs typically have key roles in governing lineage-specific expression patterns (*Nutt and Kee, 2007*; *Ludwig et al., 2019*), we identified a number of factors that demonstrate biases in expression across lineages, and for which motif occurrence is enriched in putative lineage-biased regulatory elements. Finding that a simple model of a GRN was insufficient to explain the behaviors observed in our epigenetic and expression datasets, we applied a per-locus approach to integrating our ATAC- and RNA-seq genes, defining sets of genes with different putative regulatory behaviors. We found that these groups exhibited different patterns of differential gene expression, associations with chromatin accessibility data, and, importantly, the enriched occurrence of lineage-biased TFs (*Figure 4*). Notably, our finding that grouped genes differed in their degree of differential expression is consistent with a previous study that stratified immune genes on the basis of regulatory behaviors (*Yoshida et al., 2019*). We leveraged these findings to identify TFs exhibiting enrichments for DARs around DEGs we defined as either 'enhancer-centric' or 'combo-centric' enrichments exclusive to a particular lineage (*Figure 5*). This approach pinpointed a subset of TFs whose binding motifs are significantly enriched in the corresponding chondrogenic lineage (discussed below), of which we functionally tested *RELA* and *RUNX2*. *RELA*, also known as p65, belongs to the NF-κB family of TFs that share a REL homology domain and can form transcriptionally active dimers with other family members. It is a transcriptional activator of *SOX9*, a master regulator of chondrocyte differentiation, as well as early differentiation and anabolic factors such as *SOX6* and *COL2A1*, late-stage factor *HIF-2a*, and the catabolic gene *ADAMTS5*. It also plays a role in cartilage homeostasis (*Yu et al., 2020*) and degradation in osteoarthritis (*Zhao et al., 2020*; *Olivotto et al., 2015*; *Kobayashi et al., 2016*; *Saito et al., 2010*; *Kobayashi et al., 2013*; *Ushita et al., 2009*). *RUNX2*, also known as *CBFA1*, *PEBP2*, or *AML3*, belongs to a class of TFs containing a Runt-homology domain (*Ogawa et al., 1993*). *RUNX2* has long been recognized as a 'master' skeletogenic factor, sitting atop a regulatory cascade governing osteoblast differentiation (*Komori et al., 1997*; *Ducy et al., 1997*; *Otto et al., 1997*). Since its initial discovery, the role of *RUNX2* in skeletogenesis has expanded to include the regulation of chondrocyte hypertrophy in growth plate cartilage (*Inada et al., 1999*; *Kim et al., 1999*; *Yoshida et al., 2004*). It also has a similar, though pathogenic, role in articular chondrocytes, which acquire hallmarks of hypertrophy in joint diseases such as osteoarthritis (*Chen et al., 2020*; *Catheline et al., 2019*; *Liao et al., 2017*). Remarkably, when we performed ChIP-qPCR against candidate regulatory regions with RELA or RUNX2 binding sites, we found that a majority (16 of the 17 tested) did in fact bind the predicted TF in the expected lineage. These findings emphasize the co-use of the epigenetic and expression datasets generated in this study in defining putative gene regulatory networks which may be active in developing chondrocyte populations, and in identifying key regulatory factors controlling these networks.

Our transcriptomic profiling approach uncovered additional lineage-specific TFs in both hESC-derived cartilages, including those that were identified as differentially expressed in the larger group of samples and in the developmental in vitro timecourse, but not in the smaller subset of samples (batch 2) in which we also performed ATAC-seq. Many of these TFs and their associated family members exhibited similar expression patterns in the fetal cartilage specimens, and many have been previously identified in the context of cartilage and joint biology, once again validating the hESC-model system we've established. Such TF families identified in the TGFB-induced hESC-derived articular cartilage include the ETS factors, containing a conserved ETS DNA-binding domain, including the

polyomavirus enhancer activator 3 (*PEA3*) family members (*ETV1, ETV4, ETV5*), and the ETS-related gene (*ERG*) family members (*ERG, FLI1, FEV*) (*Findlay et al., 2013*). Here, we specifically pinpointed *ETV1* and *FLI1* as regulators of enhancer- and combo-centric genes in the hESC-derived articular cartilage lineage (*Figure 5*). *PEA3* family members are significantly differentially expressed in both hESC-derived articular cartilage and in fetal epiphyseal chondrocytes. They are FGF-responsive genes and there is some evidence that loss of these proteins results in reduced and disorganized brachial cartilage (*Herriges et al., 2015*). *ERG* and *FLI1* are differentially expressed in hESC-derived articular cartilage (but not significant in fetal data), while *FEV* is differentially expressed in growth plate cartilage (not in fetal). *ERG* has a role in the long-term maintenance of articular cartilage, and, along with *FLI1*, regulates articular cartilage genes such as *PTHLH* and *PRG4* (*Iwamoto et al., 2007*; *Larmour et al., 2013*). The CREB family of TFs includes *CREB5* and *CREB3L1*, both of which are differentially expressed in hESC-derived articular cartilage (CREB5 is also differentially expressed in fetal epiphyseal cartilage). *CREB5* is a known regulator of *PRG4* expression in articular cartilage (*Zhang et al., 2021*), and shares sequence homology with the ATF family of TFs, such as *ATF7* which we highlight as a regulator of combo-centric genes in the hESC-derived articular cartilage lineage and which has near-identical DNA binding motif compared to *CREB5* (*Figure 5*). Nuclear factor of activated T-cells (NFAT) family members *NFATC2* and *NFATC4* are both differentially expressed in hESC-derived articular cartilage (*NFATC4* is also differentially expressed in the fetal epiphysis). *NFATC2* is also more highly expressed in superficial zone chondrocytes compared to deep zone chondrocytes in bovine cartilage (*Zhang et al., 2021*), and NFAT family members play a role in chondrocyte gene expression and articular cartilage maintenance (*Tomita et al., 2002*; *Greenblatt et al., 2013*; *Tardif et al., 2013*). The homeobox proteins *MEOX1* and *MEOX2* and the LIM-homeobox protein *LHX9* were also DEGs in both hESC-derived articular cartilage and fetal epiphyseal chondrocytes. *MEOX1* and *MEOX2* are essential for the development of all somite compartments and for the normal development of the craniocervical joint (*Skuntz et al., 2009*). LHX9 is induced by FGF-signaling and has been previously studied for its role in the progression of osteosarcomas (*Li et al., 2019*). These TFs and TF families, among others we identified in these studies, warrant further exploration of their individual and joint roles in articular cartilage development and stability.

In the growth plate lineage, members of the DLX family of TFs, *DLX2, DLX5, and DLX6* were highly expressed in growth plate cartilage (*DLX5* and *DLX6* were also differentially expressed in the fetal growth plate), and are known to be critical regulators of cartilage differentiation during endochondral ossification. In particular, *DLX5* has been shown to regulate the differentiation of immature proliferating chondrocytes into hypertrophic chondrocytes, and in osteoblast differentiation (*Ferrari and Kosher, 2002*). Similarly, two *RUNX* family members, *RUNX2* and *RUNX3* are differentially expressed in both hESC-derived and fetal growth plate cartilage. *RUNX2*, as discussed above, is a critical TF for chondrogenic maturation and osteoblast differentiation, and can cooperate with *DLX5* and *SP7* for the proper skeletal development (*Komori, 2015*). *RUNX3* works redundantly with *RUNX2* in chondrocyte maturation (*Yoshida et al., 2004*). The forkhead box (FOX) proteins are a superfamily of TFs, of which several members are differentially expressed in either articular cartilage or growth plate lineages. Of this large family, *FOXA2*, expressed in the hESC-derived growth plate cartilage, is a critical regulator of hypertrophic differentiation in chondrocytes and has been implicated in cartilage degradation and OA progression (*Ho et al., 2019*). *Myocyte enhancer factor 2* C (*MEF2C*) is differentially expressed in both hESC-derived and fetal growth plate cartilage and activates the genetic program for hypertrophy during endochondral ossification (*Arnold et al., 2007*). These TFs, and others identified in the studies herein, can now be investigated for their biological role in growth plate biology and chondrocyte function.

The molecular data provided herein have, for the first time, unlocked key findings regarding human articular and growth plate cartilage development. We established and validated our in vitro human pluripotent stem cell cartilage differentiation system as a predictive tool in investigating articular and growth plate cartilage lineages. This is particularly important for understanding how to specify and maintain articular cartilage, since diseased and sometimes even regenerating tissue following cartilage damage display hypertrophy-like changes (*van der Kraan and van den Berg, 2012*). Novel genes that are expressed differentially between the two different tissues were also identified, some of which exhibit zone-specific expression patterns within developing cartilage. Continued efforts to identify genes and networks that regulate cartilage development will undoubtedly be propelled by

these comprehensive comparative analyses of transcriptomic and epigenetic signatures of human articular and growth plate cartilage.

# Methods

**Key resources table**

| Reagent type (species) or resource | Designation | Source or reference | Identifiers | Additional information |
|---|---|---|---|---|
| Cell line (human) | H9 hESCs (XX) | Wicell | WAe009-A | |
| Biological sample (human) | Musculoskeletal/joint fetal donor samples; from first trimester termination | Birth Defects Research Laboratory, University of Washington | | |
| Biological sample (human) | Phalangeal joint fetal donor samples; first trimester termination (E70) | Advanced Bioscience Resources Inc. | | |
| Antibody | Rabbit polyclonal anti-EFHD1 | Sigma Aldrich | Cat# HPA056959 | 1:100 |
| Antibody | Rabbit polyclonal anti-COL15A1 | Sigma Aldrich | Cat# HPA017915 | 1:100 |
| peptide, recombinant protein | Basic Fibroblast Growth Factor (bFGF) | R&D Systems | Cat#233-FB | |
| Peptide, recombinant protein | Bone Morphogenetic Protein 4 (BMP4) | R&D Systems | Cat#314 BP | |
| Peptide, recombinant protein | Activin A | R&D Systems | Cat#338-AC | |
| Peptide, recombinant protein | Transforming Growth Factor Beta 3 (TGFB3) | R&D Systems | Cat#243-B3 | |
| Chemical compound, drug | Fetal Bovine Serum (FBS) | Corning | Cat#35–010-CV | |
| Chemical compound, drug | Dulbecco's Modified Eagle Medium, high glucose | Gibco | Cat#11995065 | |
| Chemical compound, drug | DMEM/F12 | Corning | Cat#10–092-CV | |
| Chemical compound, drug | StemPro-34 serum-free medium and nutrient supplement | Gibco | Cat#10639011 | |
| Chemical compound, drug | Knockout Serum Replacement (KOSR) | Gibco | Cat#10828028 | |
| Chemical compound, drug | L-Ascorbic Acid | Sigma-Aldrich | Cat#A4544 | |
| Chemical compound, drug | L-glutamine | Sigma-Aldrich | Cat#25030081 | |
| Chemical compound, drug | Non-essential amino acids (NEAA) | Gibco | Cat#11140050 | |
| Chemical compound, drug | Penicillin/Streptomycin (Pen/Strep) | Gibco | Cat#15140122 | |
| Chemical compound, drug | b-mercaptoethanol, 55 mM solution (BME) | Gibco | Cat#21985023 | |
| Chemical compound, drug | Transferrin from human serum | Roche | Cat#10652202001 | |
| Chemical compound, drug | a-monothioglycerol (MTG) | Sigma-Aldrich | Cat#M6145 | |
| Chemical compound, drug | Insulin-Transferrin-Selenium-Sodium Pyruvate (ITS-A) | Gibco | Cat#51300044 | |
| Chemical compound, drug | L-Proline | Gibco | Cat#P5607 | |

*Continued on next page*

*Continued*

| Reagent type (species) or resource | Designation | Source or reference | Identifiers | Additional information |
|---|---|---|---|---|
| Chemical compound, drug | Dexamethasone | Sigma-Aldrich | Cat#D4902 | |
| Chemical compound, drug | Polyheme (2-hydroxyethyl methacrylate) | Sigma-Alrich | Cat#P3932 | |
| Chemical compound, drug | Gelatin from porcine skin type A | Sigma-Alrich | Cat#G1890 | |
| Chemical compound, drug | Matrigel, growth factor-reduced | Corning | Cat#354230 | |
| Chemical compound, drug | TryplE | Gibco | Cat#12605028 | |
| Chemical compound, drug | Trypsin from porcine pancreas | Sigma-Aldrich | Cat#T4799 | |
| Chemical compound, drug | Ethylenediaminetetraacetic Acid (EDTA), 0.5 M solution, pH 8.0 | Sigma-Aldrich | Cat#A3145 | |
| Chemical compound, drug | DNaseI from bovine pancreas | Sigma-Aldrich | Cat#260913 | |
| Chemical compound, drug | Collagenase type B | Roche | Cat#11088831001 | |
| Chemical compound, drug | ROCK inhibitor Y-27632 dihydrochloride (RI) | Tocris | Cat#1254 | |
| Chemical compound, drug | SB431542 hydrate | Sigma-Aldrich | Cat#S4317 | |
| Chemical compound, drug | Dorsomorphin (DM) | Sigma-Aldrich | Cat#P5499 | |
| Chemical compound, drug | IWP2 | Tocris | Cat#3533 | |
| Chemical compound, drug | Collagenase Type I | Sigma-Aldrich | Cat#C0130 | |

## Maintenance of hESCs

All reported research involving human embryonic stem cells was approved by IRB (IRB-P00017303) and ESCRO (ESCRO-2015.4.24) regulatory bodies at Boston Children's Hospital. H9 hESCs (Wicell, RRID:CVCL_9773) were maintained on irradiated mouse embryonic fibroblasts in hESC media containing DMEM/F12 (Corning) supplemented with 20% knockout serum replacement (Gibco), nonessential amino acids (Gibco), L-glutamine (Gibco), Pen/Strep (Gibco), b-mercaptoethanol (Gibco), and human bFGF (10 ng/mL) in six-well tissue culture plates. Cells were passaged when they reached ~80% confluency onto new feeders as cell clusters of about 3–10 cells, following dissociation with TrypLE (Gibco). The identity of the H9 hESC line was authenticated at the commercial source (Wicell) using STR and karyotype and was mycoplasma negative.

## Generation of chondrocytes from hESCs

A detailed description of the protocol for generating chondrogenic cells and tissues from human pluripotent stem cells has been published (*Craft et al., 2015*). Briefly, embryoid bodies (EBs) were generated from H9 hESCs and cultured in suspension in the presence of BMP4 (1 ng/mL) and ROCK inhibitor (5 µM) for 24 hr in StemPro-34 medium (Gibco) supplemented with L-glutamine (Gibco), L-ascorbic acid (Sigma-Aldrich), transferrin (Roche), and a-monothioglycerol (Sigma-Aldrich). On day 1, EBs were harvested and resuspended in StemPro-34 media with bFGF (5 ng/mL), BMP4 (3 ng/mL), Activin A (2 ng/mL), and ROCK inhibitor (5 µM) to induce primitive streak-like mesoderm. After 44 hr, on day 3, the EBs were harvested from the induction media, cells were dissociated with TrypLE and cultured as monolayers (100,00 cells per well) in 96-well tissue culture plates (Corning) in StemPro-34 media containing bFGF (20 ng/mL), an inhibitor of type I activin receptor-like kinase (ALK) receptors

SB431542 (5.4 µM), type I BMPR inhibitor dorsomorphin (4 µM), and a Wnt inhibitor IWP2 (2 µM). After 48 hr, on day 5, monolayer cultures were maintained in StemPro-34 media containing bFGF (20 ng/mL) until day 14 to generate chondrogenic mesoderm. Cultures were maintained in a hypoxic 5% O2, 5% CO2, 90% N2 environment for 11 days, and normoxic 5% CO2/air condition for the remainder of the culture period.

Cartilage tissues were generated from the hESC-derived chondrogenic mesoderm culture on day 14 by plating cells in micromass culture. Briefly, 250,000 cells were seeded onto 24-well tissue culture plates (Corning) coated with Matrigel (Corning) in base chondrogenic media consisting of high glucose DMEM (Gibco) supplemented with 1% ITS-A, L-proline (40 µg/ml) (Sigma-Aldrich), dexamethasone (0.1 µM) (Sigma-Aldrich), L- ascorbic acid (100 µg/mL) and TGFβ3 (10 ng/mL) for 2 weeks to generate chondroprogenitors. They were then maintained for an additional 10 weeks in TGFβ3 (10 ng/mL) to generate articular cartilage tissue or transitioned to base chondrogenic media containing L-ascorbic acid (100 µg/mL) and BMP4 (50 ng/mL) to generate growth plate-like cartilage. Cells and/or tissues were collected after 4, 8, or 12 weeks in micromass.

Chondrocytes were generated for the transcriptomic and epigenetic studies in seven independent experiments. For each cell type (articular and growth plate), a single micromass was collected per replicate, as described in *Supplementary file 1a and 2a*. Additional experiments were performed to produce cartilage tissues for validation.

## Fetal tissue dissection

Human fetal donor samples (E59, E67, E72) were collected from the first trimester termination via the University of Washington (UW) Birth Defects Research Laboratory (BRDL) in full compliance with the ethical guidelines of the NIH and with the approval of UW Review Boards for the collection and distribution of human tissue for research, and Harvard University and Boston Children's Hospital for the receipt and use of such materials (Capellini: IRB16-1504; Craft: IRB-P00017303). The samples were briefly washed in Hank's Balanced Salt Solution (HBSS) and transported in the same buffer at 4 °C during shipment.

Cartilaginous tissues as described below were dissected under a light dissection microscope in 1 x phosphate-buffered saline (PBS) on ice and soft tissues were removed. Where appropriate, each epiphysis or growth plate chondrocyte population was microdissected and cells were isolated by collagenase treatment independently.

For fetal bulk-RNA seq, epiphyseal, and growth plate cartilages from the left and right distal femur were micro-dissected from an E67 donor sample. For qPCR, epiphyseal and growth plate cartilage from the distal femur and proximal tibia of one knee joint were microdissected from E59, E67, and E72 human samples. Donor samples for histology/immunohistochemistry were obtained from E59 knee joints (UW BDRL) and E70 metacarpophalangeal and metatarsophalangeal joints (Advanced Bioscience Resources Inc), formalin-fixed and paraffin embedded as described in the Histology/Immunohistochemistry section.

## RNA-sequencing (RNA-seq)

Cartilage derived from hESCs (Batch 1 and 4) was enzymatically digested with 0.2% type I collagenase (Sigma, St. Louis) for up to 2 hr at 37 °C to solubilize the majority of ECM. Microdissected fetal cartilage was minced and subsequently incubated with 0.1% bacterial collagenase (Sigma, St. Louis) for 2–3 hr at 37 °C to solubilize the majority of ECM. Liberated cells were pelleted by centrifugation and the supernatant was completely removed. Fetal chondrocytes and hESC-derived cells from Batch 1/4 were lysed in guanidine thiocyanate buffer. Total RNA was purified using silica column-based kits (ThermoFisher). RNA quality and quantity were assessed via Bioanalyzer (Agilent, Santa Clara), with RIN values >7. 100 ng of total RNA was used as input for the TruSeq RNA Library Prep Kit v2 (Illumina, San Diego). Libraries for Batch 4 were prepared using the NEBNext Ultra II RNA Library Prep Kit for Illumina using the manufacturer's instructions (Azenta). Libraries from Batch 1 were sequenced on an Illumina NextSeq instrument using 2 × 150 bp paired-end reads. The fetal samples (Batch 3) plus one technical replicate of an hESC-derived articular and growth plate library from Batch 1, and Batch 4, were sequenced on an Illumina Hi-seq using 2 × 150 bp paired-end reads. hESC-derived cartilage tissues from Batch 2 were lysed directly in Trizol (Thermo Fisher, Waltham) without prior

ECM dissociation (see Paired ATAC-seq and RNA-seq section below). See *Supplementary file 1a* for additional details.

## Paired ATAC-seq and RNA-seq

In order to perform simultaneous RNA-seq and ATAC-seq assays on the same micromass chondrocyte culture, we physically bisected cultures via forceps and a scalpel blade. One section of bisected micromass culture was immediately minced, then subsequently transferred to a 1.5 ml microfuge tube containing 200 uL of TRIzol reagent (Thermo Fisher Scientific, Waltham) and 5 mm stainless steel beads (Qiagen). Sections were homogenized using a tissue lyser (Qiagen) at 50 Hz for 2 min periods with intermittent incubation on ice for 1 min. RNA was extracted from the tissue homogenate using the phenol-chloroform extraction method, followed by purification using the Direct-zol RNA miniprep kit (Zymo Research, Irvine). RNA quality and quantity was assessed using Bioanalyzer (Agilent, Santa Clara) and Qubit (Thermofisher, Waltham). Samples with RNA integrity numbers >7 were used for subsequent RNA-seq experiments. 100 ng of total RNA were then sent to the Harvard University Bauer Core Facility for library preparation with the Kapa mRNA Library Prep kit (Roche, Basel). Generated RNA-seq libraries were then sequenced on a single sequencing lane of the Illumina NextSeq 500 using 2 × 38 bp reads at the Harvard University Bauer Core Facility – see *Supplementary file 1a* for per-sample sequencing information. Sequencing yielded ~500 million reads per lane and an average of 50 million per sample. Quality control statistics and primer information are presented in *Supplementary file 1a*.

The second section of micromass culture was again bisected, then transferred to a 1.5 ml DNA LoBind tube (Eppendorf) containing 200 µl of DMEM media supplemented with 5% FBS. To generate a single-cell suspension, each sample was then subjected to 1% collagenase, type 2 (Worthington Biochemical, New Jersey) digestion for 2 hr at 37 °C rocking, mixing every 30 min. Following digestion, the suspension was vortexed, then subsequently centrifuged at 500 × g at 4 °C for 5 min. Pellets were resuspended in 200 µL 5% FBS/DMEM for cell counting. All cell counting methods were performed using trypan blue and a hemocytometer and subsequent ATAC-seq steps were performed on those samples that had cell death rates well below 10%. Next, cells were re-suspended in concentrations of 50,000 cells in 1 x PBS. Cell samples were then subjected to the ATAC-seq protocol as described previously (*Buenrostro et al., 2015*; *Buenrostro et al., 2013*), modifying the protocol by using 2 µl of transposase per reaction.

The transposase reaction product was then purified using the DNA Clean & Concentrator Kits (Zymo Research) following manufacturer's protocols, eluted in 10 µl of warmed ddH20, and stored at −20 °C. All samples were next subjected to PCR amplification and barcoding following *Buenrostro et al., 2015*; *Buenrostro et al., 2013*. Ten microliters of transposed DNA were then placed in a reaction containing NEBNext High-Fidelity PCR Master Mix, ddH20, and barcoding primers. Following PCR amplification, samples were subjected to double-sided size selection using the Magbind RXN Pure Plus beads (OMEGA) following manufacturer's instructions. The samples were eluted in 20 µl of TE buffer, nano-dropped, and the fragment size distribution was analyzed by the 2100 Bioanalyzer instrument (Agilent Technologies). Prior to sequencing, library concentrations were determined using the KAPA Library Quantification Complete Kit (KK4824). Samples were then sent out to the Harvard University Bauer Core Facility for sequencing on a single lane of an Illumina NextSeq 500. Quality control statistics and primer information are presented in *Supplementary file 3a*.

## Quantitative RT-PCR

Total RNA was extracted from in vitro tissue using the MagMAX mirVana Total RNA kit (Applied Biosystems) and from fetal tissue using the RNAqueous-Micro Total RNA kit (Invitrogen). RNA (0.1–1 µg) was reverse transcribed with Superscript IV VILO reverse transcriptase (Invitrogen) and treated with ezDNase enzyme (Invitrogen). Real-time quantitative PCR was performed on a ViiA 7 Real-Time PCR System with OptiFlex Optics System (Applied Biosystems) using PowerUp SYBR Green PCR kit (Applied Biosystems). Genomic DNA standards were used to evaluate the efficiency of the PCR and calculate the copy number of each gene relative to the expression of the gene encoding TATA-box binding protein (*TBP*). All data represent three biological replicates (independent experiments or fetal donor specimens) or more as indicated. Student's t-test was used to evaluate statistical significance, as indicated. Oligonucleotides are provided in *Supplementary file 6*.

## Histology and immunohistochemistry

In vitro-derived cartilage tissues and primary human donor samples (E59 knee joint; E70 metacarpo-phalangeal and metatarsophalangeal joints) were fixed in 10% formalin and embedded in paraffin. 5 µm sections were stained with toluidine blue to visualize sulfated glycosaminoglycans. Immunohistochemistry was performed using antibodies recognizing EF-Hand domain family D1 (HPA056959; RRID:AB_2683288; Sigma Aldrich), and type XV collagen (HPA017915; RRID:AB_1847100; Sigma Aldrich). Antigen retrieval was performed on the tissue using citrate buffer (pH 6.0, overnight at 50 °C) for type XV collagen, and using pepsin (30 min. at 37 °C) for EFHD1. Positive staining was visualized with DAB. Sections were counterstained with Mayer's hematoxylin (blue).

## RNA scope

Five µm sections of tissues were deparaffinized by submersion in xylene and washed with 100% ethanol, and treated with the RNAscope hydrogen peroxide solution (Advanced Cell Diagnostics (ACD), Cat. No. 322381). Target antigen retrieval was performed by incubating sections in TEG buffer (25 mM Tris-HCl pH 8, 10 mM EDTA, and 50 mM Glucose) at 60 °C for 4 hr, changing the buffer every 40 min. Sections were then rinsed in water, dipped in 100% ethanol for 3 min, and air dried. RNAscope Protease 3 was applied to the sections at 40 °C for 1 hr (ACD, Cat. No. 322381). In situ detection of *PRG4*, *TNMD, COL2A1,* and *COL10A1* mRNA was performed using the RNAscope Multiplex Fluorescent V2 Assay system (ACD, Cat. No. 323110) with the probes (Probe-Hs-COL2A1, Cat. No. 427878; Probe-Hs-PRG4-C3, Cat. No. 427861-C3; Probe-Hs-TNMD, Cat. No. 564409; Probe-Hs-COL10A1, Cat No. 427851). Fluorescent signal was detected using a confocal microscope (Zeiss LSM 800).

## RNA-seq processing

Paired-end sequencing reads were mapped to the human reference transcriptome (GRCh37.67) obtained from the ENSEMBL database using the 'quant' function of Salmon version 0.14.0 (*Patro et al., 2017*) (version 1.2.1 for the timecourse) with the following parameters: '-l A --numBootstraps 100 --gcBias --validateMappings,' all others parameters were left to defaults. Salmon quantification files were imported into R version 3.6.1 (*R Development Core Team, 2023*) using the tximport library (version 1.14.0) (*Soneson et al., 2015*) with the 'type' option set to 'salmon,' all others set to default. Salmon quantification files for the timecourse were imported into R version 4.2.1 using the tximport library (version 1.24.0). Transcript counts were summarized at the gene level using the corresponding transcriptome GTF file mappings obtained from ENSEMBL.

Count data was subsequently loaded into DESeq2 (*Love et al., 2014*) version 1.26.0 (version 1.36.0 for the time course) using the 'DESeqDataSetFromTximport' function. For differential-expression analysis, a low-count filter was applied prior to normalization, wherein a gene must have had a quantified transcript count greater than five in at least three samples in order to be retained. PCA of samples across genes was done using the 'vst' function in DESeq2 with default settings and was subsequently plotted with ggplot2.

Statistical analysis was performed using the 'DESeq' function of DESeq2 using all samples, with results subsequently summarized using the 'results' function for the BMP-TGF contrast with the 'alpha' parameter set to 0.05; p-values were adjusted using the Benjamini-Hochberg FDR method (*Benjamini and Hochberg, 1995*), with DEGs defined at an adjusted p-value cutoff of 0.05. To visualize the expression patterns of these genes across tissues, count data was normalized using the 'estimateSizeFactors' function using the default 'median ratio method,' with the normalized matrix output using the 'counts' function with the 'normalized' option. The top 200 DEGs (sorted by absolute logFC) were subset from this normalized matrix, z-score transformed using the 'scale' function in base R, and visualized with the ComplexHeatmap package version 2.4.3 (*Gu et al., 2016*) (timecourse data visualized with gplots package version 3.1.3).

## Note on digestion protocol for extracting RNA from cartilage

For transcriptomic analysis of hESC-derived chondrocytes, we isolated RNA from half the samples after first separating the cells from their extracellular matrix via enzymatic digestion (collagenase). From the remaining samples, we isolated RNA directly without prior disruption of the matrix. Differential expression analysis revealed a subset of genes up-regulated in collagenase-digested cells, which were substantially enriched for GO biological processes such as 'response to lipopolysaccharides,'

'response to oxidative stress,' and 'cellular response to toxic substance.' Several of these processes were previously identified via a similar analysis of collagenase-digested osteoblasts (*Ayturk et al., 2013*). Various 'cellular responses' were among the enriched processes, indicating chondrocytes are sensitive to enzymatic disruption of their matrix. For applications in which matrix disruption is unavoidable, such as single-cell RNA-seq, care should be taken to ensure the transcriptomic signatures of interest are not confused with artifacts associated with enzymatic digestion.

## Shared direction analysis

In order to compare the direction of differential expression between in-vitro and in-vivo samples, we took the in-vitro dataset and subset the top 100 most strongly up-regulated and down-regulated genes, for a final set of 200 (see *Supplementary file 1*). The equivalent data for these genes were subsets from the in-vivo dataset. The resulting sets of log2 fold-change values were plotted using ggplot2. A chi-square test for shared direction was performed in base R using a 2 × 2 contingency table of values (see *Supplementary file 1*). An equivalent analysis was performed using 200 genes defined in the in-vivo dataset, see *Figure 1—figure supplement 2*, *Supplementary file 1*.

## Definition of differentially-expressed TFs

To identify TFs that are DEGs in our transcriptional data, we extracted all defined motif position-weight-matrices (PWMs) from the JASPAR 2020 database (*Mathelier et al., 2016*) along with the motif database provided by HOMER version 4.11 (*Heinz et al., 2010*). We then intersected these sets of genes with those differentially expressed in our RNA-seq analysis to define our sets of differentially-expressed transcription factors (*Supplementary file 1e*). To identify differentially expressed transcription factors in our timecourse transcriptional data, we intersected the DEGs with the list of curated TFs identified in *Lambert et al., 2018*; *Supplementary file 2e-g*. For our analyses integrating ATAC and RNA-seq datasets, we defined differentially-expressed transcription factors using the expression data from the RNA-seq for which paired ATAC was generated (i.e. 'Batch 2' samples).

## Gene-set enrichment analyses

Genes associated with different expression/accessibility sets (as described in the results and methods text) were tested for enrichment in GO Biological Process terms using the 'enrichGO' function from the clusterProfiler (*Yu et al., 2012*) package version 3.13.1 (version 4.4.4 for the timecourse). The background gene sets used for individual enrichment tests are specific - pertaining to a particular analysis. For all differential-expression datasets, all genes in the human reference transcriptome (GRCh37.67), following quality filtering (see RNA-seq section of Methods), were used. For GO enrichments on defined differentially-accessible promoter windows, the background gene set was defined as the set of all promoters. Semantically similar enriched GO terms were subsequently collapsed using the 'simplify' function from clusterProfiler, using default settings. The top enriched GO terms (sorted by adjusted p-value) for each region-associated gene set are reported in *Supplementary file 1c and 2d*, limiting to the top twenty significant (adjusted p-value <0.05) terms.

## ATAC-seq read processing

Sequence read quality was checked with FastQC and subsequently aligned to the human reference hg19 genome assembly with Bowtie2 v2.3.2 (*Langmead and Salzberg, 2012*) using default parameters for paired-end alignment. Reads were filtered for duplicates using picard (https://github.com/broadinstitute/picard; version 2.18.12; RRID:SCR_006525): and subsequently used for peak calling using MACS2 (*Zhang et al., 2008*) software (version 2.1.1.2) with the following flags: 'bampe call -f BAMPE –nolambda'. Reproducible called peaks were defined using an IDR threshold of <0.05, as defined by the IDR statistical test (*Li et al., 2011*) (version 2.0.3), as well as a more stringent cutoff of 0.01 for peak sets containing more than 100,000 peaks as recommended by previous ENCODE processing pipelines. Briefly, the IDR method looks for overlaps in peak calls across pairs of replicate samples by comparing ranked peak lists (using MACS2 q-value) to define a reproducibility score curve. All paired ranks are assigned a pointwise score based on this curve, subsequently sorted, and all peaks falling below an 'irreproducible discovery rate' (IDR) threshold of 0.05 are taken as our final reproducible peak set.

For calculating differential accessibility of regions between treatments, all IDR-filtered peaks from both sets of treated tissue fragments (BMP and TGF) were padded to a fixed size of 1000 bp (from called peak centres) and pooled using bedtools (*Quinlan and Hall, 2010*) version 2.29.1. For each pooled peak, a 1000 bp window was defined (500 bp up/downstream of peak centre); along this window, a 250 bp sliding window (chosen based on the averaged called peak size across individual IDR-filtered peaks from each set) was slid in 50 bp increments to generate a set of overlapping regions (using the R language version 3.6.1). ATAC-seq read coverage within these regions was calculated using the 'bedcov' function of samtools (*Li et al., 2009*) version 1.5 for each mapped.bam file corresponding to individual ATAC-seq samples. This sliding-window approach is used to identify the area of strongest cross-sample signal (i.e. ATAC-seq read coverage) around a pooled ATAC-seq peak, improving our confidence in defining differential-accessibility (i.e. avoiding regions around an ATAC-seq peak with lower read coverage, which may make our analysis more sensitive to noise). For sets of overlapping windows corresponding to a single ATAC-seq peak, we applied a smoothening algorithm to eliminate extreme values occurring in overlapping windows for individual samples. Windows whose read coverage fell outside one standard deviation of the set (of sliding windows for a single peak) were assigned the average read coverage of the two windows adjacent. This adjustment does not impact the later differential-accessibility analysis, for which we take the original read coverage values calculated for a given window using the mapped bam file. Rather, this will impact the choice of sliding window assigned to represent this given peak, and is done to avoid consistently using the most-extreme windows to define differential accessibility (which may otherwise bias our results).

Following this smoothening, for each window, we calculated the 75th percentile read coverage value across all pelvic elements and samples (this was found to be more robust than mean read coverages, even after smoothening adjustment) using the dplyr (*CRAN, 2022*) package version 1.0.7. The window with the greatest 75th-percentile coverage was then selected as the representative region for that pooled ATAC-seq peak. Subsequently, raw read coverages for all optimized windows across all ATAC-seq samples were imported as a matrix into DESEQ2 version 1.26.0 using the 'DESeqDataSetFromMatrix' function, with differential-accessibility calculated using the 'DESeq' function with treatment-type as the main variable. Differentially accessible (DA) peaks were generated using the 'results' function from DESeq2 using the BMP-TGF contrast, with significance assessed as a Benjamini-Hochberg adjusted p-value of <0.05. To visualize accessibility in these DA peaks, read coverages were normalized using the 'estimateSizeFactors' function in DESeq2 using the default TMM normalization. Subsequently, the normalized read-counts matrix was z-score transformed using the 'scale' function in base R and plotted using the 'ComplexHeatmap' package version 2.4.3.

## De-novo motif enrichments

Sequence sets for the sets of differentially-accessible regions (as defined above) were generated using reference sequences from hg19. HOMER (version 4.11) de novo motif analysis was performed on each sequence set using a 10 x random shuffling as a background set. De novo motifs were compared to a vertebrate motif library included with HOMER, which incorporates the JASPAR database (2022 version); matches are scored using Pearson's correlation coefficient of vectorized motif matrices (PWMs), with neutral frequencies (0.25) substituted for non-overlapping (e.g. gapped) positions. Best-matching motif PWMs for TGF- and BMP-biased region sets are shown in *Figure 2C* and *Supplementary file 3d-e*.

## Targeted motif enrichments

We took our set of differentially-expressed transcription factors up-regulated in articular and growth-plate chondrocytes, defined using expression from Batch 2 samples, and obtained the PWMs for all factors. In instances where a factor had more than one defined PWM (e.g. due to overlap between JASPAR and HOMER databases), the information content (IC) of each matrix was calculated, with the matrix having the highest IC retained. To define the background nucleotide frequencies of our ATAC-seq region sets, we took the pooled set of fixed-size peaks used in the DA analysis (n=37780) and generated a Markov background model using the 'fasta-get-markov' function from the MEME suite (*Bailey et al., 2009*; version 5.4.1). Next, we took the sets of nucleotide sequences corresponding to DA peaks and scanned them for instances of particular TF motifs using the FIMO program from MEME, defining a p-value threshold for motif hits as 2e−4 (calculated as 0.1/ (2 × 250 bp of sequence)). To

define a background expectation of motif hits for sequence sets equivalent to our target sets, we used the 'shuffle' command from bedtools, randomly shuffling our sequence set across the hg19 genome (exclusive of the true set of regions). For each randomly-shuffled set, we then scanned for motif hits with FIMO, performing this random shuffling and scanning for n=100 randomized sets. This was used to establish a random background distribution of expected motif hits for a given TF motif. Motif hit values were standardized and statistical significance was assessed using a CDF of the standard normal distribution as implemented in the 'pnorm' function in base R. For each DA sequence set, P-values for significant deviations from the background distribution were corrected for the number of TFss tested (n=194) (*Supplementary file 3e*). As further confirmation, for the top 10 most strongly-DE TFs in each lineage, we increased the number of randomly-shuffled background sets to n=1000,, finding that our enrichments were robust to the size of the background distribution.

We next checked to confirm whether these motif enrichments are reflective of the complexity of PWMs, i.e., that TFs with lower sequence binding specificity result in more promiscuous motif hits. For a given TF, we compared the fraction of all sequences in a given DA set that had at least one motif hit to the IC of the factor's PWM. These values were plotted across all TFs tested using the 'ggscatter' function from ggpubr version 0.4.0, with Pearson correlation calculated with this function (see *Figure 3—figure supplement 2*).

## GREAT analysis

GREAT (*McLean et al., 2010*) takes an input set of genomic regions along with a defined ontology of gene annotations; first, it defines regulatory domains for all genes genome-wide, then measures the fraction of the genome covered by the regulatory domains of genes associated with a particular annotation (e.g. 'cartilage development'). These fractions are used as the expectation in a binomial test counting the number of input genomic regions falling within a given set of regulatory domains, which results in the reported significance of association between an input region set and a particular gene ontology term. GREAT also performs a more traditional gene-based hypergeometric test to test for the significance of region set-ontology association. The program returns a set of enriched ontologies sorted by the joint rankings of FDR-corrected binomial and hypergeometric tests, as reported here in *Supplementary file 3c*.

## Defining promoter-accessibility

All hg19 Refseq gene TSS were obtained from the UCSC genome browser (*Karolchik et al., 2014*) and padded 5 kb up/downstream to define regions around each promoter. For each promoter, 2 kb windows were slid along this region in 50 bp increments, with per-window ATAC-seq read coverage for all samples calculated using the 'bedcov' function of samtools (version 1.5) for each mapped. bam file. A similar smoothening method as that described above for per-peak windowed accessibility was also performed here using sliding promoter window accessibility. As above, this smoothening method does not impact the later differential-accessibility analysis, for which we take the original read coverage values calculated for a given window using the mapped bam file. Following smoothing, the window with the greatest 75$^{th}$-percentile coverage was selected as the representative read coverage metric for a given promoter. Following per-promoter window selection, the final matrix of read coverages for all promoter windows across all samples was loaded into DESeq2 version 1.26.0 using the resulting in a final matrix of read coverages for all promoters across all samples. This matrix was subsequently loaded into DESeq2 version 1.26.0 using the 'DESeqDataSetFromMatrix' function, with differential accessibility calculated using the 'DESeq' function with treatment type as the main variable. DA promoters were generated using the 'results' function from DESeq2 using the BMP-TGF contrast, with significance assessed as a Benjamini-Hochberg adjusted p-value of <0.05.

## Defining cis-regulatory scores

We defined a 'cis-regulatory score' which seeks to capture information on accessibility patterns within putative regulatory elements around a given gene locus, a concept inspired by methods of integrating RNA-seq and ChIP-seq datasets (*Wang et al., 2013*). For a given gene, we collected all accessibility regions (i.e. the optimized windows described above) within 100 kb up/downstream of the TSS. Given that we also, separately, considered promotor accessibility in describing the regulatory behavior of genes (see above, and below), we explicitly excluded any regions which fell within the optimized

promoter regions defined above. For each captured accessibility region, we took the calculated fold-accessibility change (BMP-TGF contrast) and scaled it by the distance to the gene TSS using the following formula:

Per-region score = (differential-accessibility logFC) * ($e^{-(0.5 + 4 * (distance / 100kb))}$).

Where the second distance-scaling term is taken from *Tang et al., 2011*. These per-region scores were summed across all captured regions across the locus to give a single, signed lineage-specific (i.e. BMP or TGF-treated) *cis*-regulatory score. These scores were visualized per gene as an additional feature in the 'ComplexHeatmap' visualizations of expression (as seen in *Figure 2B*); blue/red scale was defined based on max/min values (respectively) for this score (positive being TGF-biased, negative being BMP-biased); points outside two standard deviations of the mean score were capped at 2*SD +/−the mean.

## Defining gene regulatory behaviors

For a given gene, all accessible elements within a 100 kb (up/downstream of the gene TSS) were collected, exclusive of the optimized promoter window (as described above). For each element, the normalized accessibility counts data (via DESeq2 using the default TMM normalization, as described above) was retrieved for all hESC samples (BMP and TGF treated). These normalized counts were subsequently scaled by distance to the TSS using the above distance-scaling term. Similarly, the normalized accessibility counts data for promoter regions (as described above) was also retrieved across all samples. Finally, RNA-seq expression data for the given gene was retrieved and normalized using the 'vst' function in varistran (*Francis Harrison, 2017*) version 1.0.4 to obtain normalized expression counts across all samples.

For this gene, three different linear regression models for RNA-seq expression were generated using the 'lm' function in base R. (1) RNA ~Promoter accessibility +Enhancer Accessibility +Promoter x Enhancer Interaction. (2) RNA ~Promoter accessibility. (3) RNA ~Enhancer accessibility. The proportion of total variance in RNA expression explained by each model was calculated by feeding individual model objects into the 'anova' function in base R and summing the individual sum-of-squares values for all model coefficients, then dividing by the total sum of squares. We took the differences between the total variance explained by the full model (1) and that explained by individual-component models (2) and (3) as an approximate measurement of the contributions of either enhancer or promoter accessibility. Additionally, we defined a metric for the additive effects of combining promoter and enhancer accessibility as the difference between the variance explained by the full model and that explained by the two models individually, while accounting for instances in which this difference is negative (i.e. when the two single-component models share overlapping information). Finally, the proportion of variance that could not be explained by the full model was also included as an additional metric.

These four per-gene metrics (contribution of enhancer information, promoter information, the combination of enhancer and promoter, and unexplained variance) were defined across all genes captured in our RNA-seq analysis. The resulting matrix of genes was subsequently visualized and clustered using the 'Heatmap' function from ComplexHeatmap version 2.8.0 using a 'k' of four (as seen in *Figure 4B*). Clustered genes were defined as being either 'unexplained,' 'combo-centric,' 'promoter-centric,' or 'enhancer-centric' based on the distribution of variance explained in the heatmap shown in *Figure 4B*. The percentage of clustered genes exhibiting significant DE between BMP and TGF-treated samples (*Figure 4C*, bottom) was calculated using the results of the differential-expression analysis above. For all genes in each cluster (regardless of significance) the absolute log2FC values were compared across clusters using the 'aov' function in base R, followed by Tukey Post-hoc correction using the TukeyHSD function in base R. Differences in absolute log2FC values were plotted using the 'plotmeans' function from gplots version 3.1.1 with default settings.

To test whether the directionality of accessibility and expression was shared in different clusters of genes we used the following approach. For each gene, the distance-scaled enhancer accessibility, promoter accessibility, and gene expression values were binarized into either 'biased towards TGF samples' or 'biased towards BMP samples.' Across all genes in a given cluster, the number of genes sharing direction in either (a) enhancer accessibility and RNA expression, or (b) promoter accessibility and RNA expression was compared to those switching direction using a chi-square test in base R. Finally, for each clustered set of genes, the RNA-seq expression, promoter accessibility, and cis-regulatory score values were visualized using ComplexHeatmap, sorting first on RNA-seq expression.

## Motif analyses with regulatory behaviors

For motif analyses in the context of our defined regulatory behaviors, we considered motif enrichments at putative regulatory regions (enhancers) around genes and gene promoter regions separately. We first took the groups of genes falling into 'enhancer-centric' and 'combo-centric' clusters, separated them into those up-regulated in articular/growth-plate chondrocytes, then defined 200 kb windows centered on the TSS of each gene. Subsequently, all DA peaks sharing directionality (i.e. articular chondrocyte DA peaks around articular chondrocyte DEGs) within these windows were collected and aggregated into a final set of sequences. For 'promoter-centric' genes, we took the set of optimized promoter regions for DEGs and aggregated them. These sequence sets were then used with the AME webserver (part of the MEME suite) (*Bailey et al., 2009*), along with the sets of differentially-expressed TF motif matrices, matching the sequence sets (i.e. articular chondrocyte-biased sequence sets were scanned with articular chondrocyte-biased TFs), using default settings. The resulting enrichments were filtered with a p-value threshold of <0.05 and summarized (see *Supplementary file 4f*).

To test the specificity of these enrichments for sequences biased towards a particular lineage, we performed an additional level of motif analysis. We collected the sets of motifs enriched for genes falling in particular regulatory groups, and differentially expressed in particular lineages, and utilized the same sequence sets pertaining to enhancers or promoters of genes. To define the background nucleotide frequencies of our ATAC-seq region sets, we took the pooled set of fixed-size peaks used in the DA analysis (n=37780) and generated a Markov background model using the 'fasta-get-markov' function from the MEME suite (*Bailey et al., 2009*). We similarly took the set of optimized promoter regions (based on the analysis described above) (n=27739) and defined a Markov background nucleotide frequency model. We then took the regulatory elements of a particular set (e.g. DA peaks biased towards articular chondrocytes, around articular chondrocyte DEGs) and scanned for instances of a given motif using the FIMO program from MEME, defining a p-value threshold for motif hits as $2e-4$ (calculated as $0.1/ (2 \times 250$ bp of sequence)). Similar to above, we defined a background expectation of motif hits for sequence sets equivalent to our target sets using the 'shuffle' command from bedtools version 2.29.1, randomly shuffling our sequence set across the hg19 genome (exclusive of the true set of regions). For scanning promoter sequences, we randomly shuffled the true set of promoters across all promoters used in the differential-promoter-accessibility analysis (n=27739), to account for background biases in the occurrence of motifs within promoter regions generally.

For each randomly-shuffled set, we then scanned for motif hits with FIMO, performing this random shuffling and scanning for n=2000 randomized sets. This was used to establish a random background distribution of expected motif hits for a given TF motif. Motif hit values were standardized and statistical significance was assessed using a CDF of the standard normal distribution as implemented in the 'pnorm' function in base R. As we sought to test the specificity of motif enrichments, all motifs which were enriched for a given regulatory set (e.g. enriched in enhancer sequences) were tested, regardless of any existing lineage bias detected in our AME analyses. This resulted in our testing n=96 TF motifs against all enhancer sequences, and n=109 TF motifs against all optimized promoter regions. For each lineage-biased sequence set (i.e. sequences associated with DEGs in articular chondrocytes or growth-plate chondrocytes), p-values for significant deviations from the background distribution were BH corrected for the number of TFs tested (either 96 or 109, based on the regulatory group of genes considered) (*Supplementary file 4g-i*).

As a more stringent analysis, we also re-ran these enrichment tests using a constrained set of randomized regions. For scanning enhancer sequences, we randomly shuffled the true target set of regions across the entire set of peaks used in the DA analysis (n=37780), to account for background biases in the occurrence of motifs across ATAC-seq peaks generally. We refer to this analysis as the 'ATAC-BACK' set. As above, 1000 randomly-generated sets were used to assess statistical significance along with BH corrections.

To visualize the differences in motif enrichments across sets of lineage-biased genes and regions, we took the sets of motifs enriched in either growth-plate (BMP) or articular (TGF) chondrocyte lineages and sorted them by the absolute difference in $\log_{10}$ adjusted p-values (i.e. selecting those motifs with the largest change in motif enrichments across lineages). The top five motifs from each analysis were then plotted using ggplot2 (as seen in *Figure 5A, B, D and E*). The 'ATAC-BACK' background sets were used to visualize the distribution of motif hits used in assessing the significance of the indicated factors in *Figure 5C and F*.

## Overlap with ChIP-seq data

For our two chosen factors of interest (RELA and RUNX2), we obtained ChIP-seq datasets from ChIP-Atlas (*Oki et al., 2018*), which aggregates ChIP-seq datasets from publicly-available datasets (see *Supplementary file 5a* for full lists of accessions aggregated in each track). We broadly selected all ChIP-seq tracks available for our factors, retaining all peaks at a significance threshold of 0.05. Tracks were sorted and merged with bedtools. To test for overlap with our region sets we performed hyper-geometric tests using the 'phyper' function in base R. Given our interest in the lineage-specificity of transcription factor motifs, we used as a background the pooled set of DA peaks biased in either lineage (n=4720). BH correction was applied for the number of sequence sets tested (n=4) (*Supplementary file 5a*). We subsequently summarized the results of these overlaps, which are presented in *Supplementary file 5b-e*. Additionally, we took the result of FIMO-predicted motif hits (as described above) for RELA and RUNX2 motif matrices on BMP- and TGF-biased peak sets, and included them along with the data on ChIP-seq overlaps. Indicated genes in *Supplementary file 5* represent the closest nearby gene to a given peak.

## Mouse chondrocyte isolation

All studies involving animals were performed in accordance with ARRIVE guidelines. All animal work was performed according to approved institutional animal care and use committee protocols at Harvard University (IACUC 13-04-161). ATAC-seq experiments were performed on transgenic Col2a1-ECFP/Col10a1-mCherry reporter mice (a gift from Dr. Cliff Tabin at Harvard Medical School) (*Choka-lingam et al., 2009*), which has an enhanced cyan fluorescent protein reporter under the control of the promoter of Col2a1, and an enhanced mCherry fluorescent protein reporter under the control of the promoter of Col10a1. Col2a1-ECFP/Col10a1-mCherry male and female mice were used to establish timed matings, and at E15.5 pregnant females were euthanized to acquire embryos. At this time point, chondrocytes are easily extracted from the surrounding extracellular matrix for ATAC-seq with negligible effects on the epigenome (*Guo et al., 2017*). Embryos were dissected under a micro-scope in 1 X PBS on ice and the proximal and distal portions of the right and left femur and tibia of the hind limb were stripped clear of soft tissues. Each proximal or distal cartilaginous end comprising of the articular chondrocytes, epiphyseal chondrocytes, and metaphyseal chondrocytes was then micro-dissected from the bony diaphysis and separately pooled from a single litter, consisting on average of eight animals. All samples were collected in micro-centrifuge Eppendorf tubes containing 200 mL 5% FBS/DMEM. To generate a single-cell chondrocyte suspension, each pooled sample was then subjected to 1% Collagenase II (Worthington Biochemical, LS004176) digestion for 2 hrs at 37 °C rocking, mixing every 30 min. After placing on ice, samples were next filtered using a micro-centrifuge filter set-up by gently mashing the residual tissues through the filter followed by rinsing with 5% FBS/DMEM. Samples were then spun down at 500 × g at 4 °C for 5 min. Col2a1-ECFP or Col10a1-mCherry positive chondrocytes were collected by fluorescence-activated cell sorting (FACS) using BD FACS Aria Cell Sorters at Harvard University Bauer Core Facility (HUBCF). All chondro-cyte counting methods were performed using trypan blue and a hemocytometer and subsequent ATAC-seq steps were performed on collected chondrocyte samples that had cell death rates well below 10%. On average we acquired 150,000–200,000 living cells for Col2a1-ECFP positive chondro-cytes per harvest; and acquired 2500–5000 living cells for Col10a1-mCherry positive chondrocytes per harvest. Next, cells were re-suspended in concentrations of 50,000 cells in 1 x PBS for CFP-positive chondrocytes; and entire mCherry-positive chondrocytes were resuspended in PBS for the next step of ATAC-seq process. Cell samples were then subjected to the ATAC-seq protocol as described previously *Buenrostro et al., 2015*; *Buenrostro et al., 2013*, modifying the protocol by using 2 μl of trans-posase per reaction. The transposase reaction product was then purified using the Omega MicroElute DNA Clean Up Kit following manufacturer's protocols, eluted in 10 μl of warmed ddH20, and stored at –20 °C. All samples were next subjected to PCR amplification and barcoding following *Buenrostro et al., 2015*; *Buenrostro et al., 2013*. Ten microliters of transposed DNA were then placed in a reaction containing NEBNext High-Fidelity PCR Master Mix, ddH20, and primers. Because of the low cell number for Col10a1-mCherry positive chondrocytes, additional PCR cycles of PCR amplification are needed to generate enough PCR fragments for the next steps of ATAC-seq. To determine the optimal number of cycles to amplify the library with minimal artifacts associated with saturation PCR of complex libraries, the appropriate number of PCR cycles is determined using qPCR to stop amplification prior

to saturation (*Buenrostro et al., 2015*; *Buenrostro et al., 2013*). Following amplification, samples were transferred to new tubes and treated using the OMEGA Bead Purification Protocol following the manufacturer's instructions. The samples were eluted in 30 ml of TE, nano-dropped, diluted to 5 ng/ml, and run on a Bioanalyzer. Prior to sequencing, sample concentrations were determined using the KAPA Library Quantification Complete Kit (KK4824). Samples were then sent out to the Harvard University Bauer Core Facility for sequencing on one lane of the Illumina NextSeq 500. Sequencing yielded ~400 million reads per lane and an average of 50 million per sample.

Sequence read quality was checked with FastQC and subsequently aligned to the mouse reference mm10 genome assembly with Bowtie2 v2.3.2 (*Langmead and Salzberg, 2012*) using default parameters for paired-end alignment. Reads were filtered for duplicates using picard (https://github.com/broadinstitute/picard; version 2.18.12; RRID:SCR_006525) and subsequently used for peak calling using MACS2 (*Zhang et al., 2008*) (version 2.1.1.2) with the following flags: 'bampe call -f BAMPE –nolambda.' Reproducible called peaks were defined using an IDR threshold of <0.05, as defined by the IDR statistical test (*Li et al., 2011*) (version 2.0.3). However, the Col10a1 datasets, for which we noted elevated levels of cell death prior to ATAC-seq (possibly attributable to the hypertrophic nature of these cells) had substantially greater variability across samples compared to our Col2a1 datasets. Thus, for Col10a1 called peaks (peak calls are available through GEO datasets), we used a less stringent approach to replicate consolidation – requiring that a called peak is overlapped in at least two different samples (using bedtools intersect) in order to be considered for subsequent analyses.

## Chromatin immunoprecipitation – quantitative PCR (ChIP-qPCR)

Chromatin preparation, ChIP, and qPCR were performed at Active Motif (Carlsbad, CA). In brief, hESC-derived articular and growth plate chondrocytes were isolated from their respective cartilage tissues, fixed with 1% formaldehyde for 15 min, and quenched with 0.125 M glycine. Chromatin was isolated by the addition of lysis buffer, followed by disruption with a Dounce homogenizer. Lysates were sonicated and the DNA sheared to an average length of 300–500 bp using the EpiShear Probe Sonicator (Active Motif, cat # 53051) with an EpiShear Cooled Sonication Platform (Active Motif, cat # 53080). Genomic DNA (Input) was prepared by treating aliquots of chromatin with RNase, proteinase K, and heat for de-crosslinking (overnight at 65 °C) followed by ethanol precipitation and SPRI bead clean up (Beckman Coulter). The resulting DNA was quantified by Clariostar (BMG Labtech). Extrapolation to the original chromatin volume allowed quantitation of the total chromatin yield.

Aliquots of chromatin (25 µg) were precleared with protein A agarose beads (Invitrogen). Genomic DNA regions of interest were isolated using antibodies against RelA (Active Motif, catalog number 39369; RRID:AB_2793231) in articular chondrocytes, and against RUNX2 (CST, catalog number 8486; RRID:AB_10949892) in growth plate chondrocytes, where the two TFs are differentially expressed, respectively. Complexes were washed, eluted from the beads with SDS buffer, and subjected to RNase and proteinase K treatment. Crosslinks were reversed by incubation overnight at 65 °C, and ChIP DNA was purified by phenol-chloroform extraction and ethanol precipitation.

Quantitative PCR (QPCR) reactions were carried out in triplicate using SYBR Green Supermix (Bio-Rad, Cat # 170–8882) on a CFX Connect Real-Time PCR system. One negative control primer pair was used (Unt12, Human negative control primer set 1, Active Motif, catalog number 71001) as well as one positive control (DPF1 for RUNX2, BIRC3 for RelA), plus the target sites of interest. The resulting signals were normalized for primer efficiency by carrying out qPCR for each primer pair using unprecipitated genomic DNA. Oligonucleotides are provided in *Supplementary file 6*.

## Acknowledgements

The authors would like to acknowledge funding sources for this work (NIAMS - R01-AR073821 (A.M.C.) and R01-AR070139 (T.D.C.)). We thank the University of Washington Birth Defects Research Laboratory, supported by NIH award number 5R24HD000836 from the Eunice Kennedy Shriver National Institute of Child Health and Human Development as well as Advanced Bioscience Resources Inc as sources for fetal donor tissue. The authors also thank Chaochang Li (Boston Children's Hospital) and Garrett Ruff (Cornell University) for imaging assistance, Evelyn Flynn (Boston Children's Hospital) for histological support, Alexander Okamoto (Harvard University) for primary tissue dissection, the Bauer Core Facility (Harvard University) for NGS support, the Massachusetts General Hospital Center for Skeletal Research NIH-funded program (P30 AR075042), and Jessica Lehoczky (Brigham and Women's

Hospital, Harvard Medical School) and Matthew Warman (Boston Children's Hospital) for critical reading of the manuscript.

## Additional information

### Funding

| Funder | Grant reference number | Author |
|---|---|---|
| National Institute of Arthritis and Musculoskeletal and Skin Diseases | R01-AR073821 | April M Craft |
| National Institute of Arthritis and Musculoskeletal and Skin Diseases | R01-AR070139 | Terence D Capellini |

The funders had no role in study design, data collection and interpretation, or the decision to submit the work for publication.

### Author contributions
Daniel Richard, Steven Pregizer, Data curation, Formal analysis, Investigation, Visualization, Methodology, Writing – original draft; Divya Venkatasubramanian, Data curation, Validation, Investigation, Writing – original draft; Rosanne M Raftery, Pushpanathan Muthuirulan, Zun Liu, Resources; Terence D Capellini, Supervision, Funding acquisition, Writing – review and editing; April M Craft, Conceptualization, Formal analysis, Supervision, Funding acquisition, Validation, Visualization, Writing – original draft, Writing – review and editing

### Author ORCIDs
Daniel Richard (ID) http://orcid.org/0000-0001-8568-9473
April M Craft (ID) http://orcid.org/0000-0002-4423-8008

### Ethics
Human subjects: Human fetal donor samples were collected from the first trimester termination via the University of Washington (UW) Birth Defects Research Laboratory (BRDL) in full compliance with the ethical guidelines of the NIH and with the approval of UW Review Boards for the collection and distribution of human tissue for research, and Harvard University and Boston Children's Hospital for the receipt and use of such materials, and Harvard University and Boston Children's Hospital for the receipt and use of such materials (Capellini: IRB16-1504; Craft: IRB-P00017303). This is not deemed human subjects research. All reported research involving human embryonic stem cells was approved by IRB (IRB-P00017303) and ESCRO (ESCRO-2015.4.24) regulatory bodies at Boston Children's Hospital. All animal work was performed according to approved institutional animal care and use committee protocols at Harvard University (IACUC 13-04-161).

### Decision letter and Author response
Decision letter https://doi.org/10.7554/eLife.79925.sa1
Author response https://doi.org/10.7554/eLife.79925.sa2

## Additional files

### Supplementary files
• Supplementary file 1. Sequencing information for RNA-seq runs. (a) Quality summary statistics for RNA-seq samples. (b) DEG data for hESC-derived (in vitro) and fetal (*primary*) chondrocyte samples. Columns C-H: DESeq2 statistics for hESC-derived samples. Log2FC is calculated as BMP-vs-TGF. Columns I-N: DESeq2 statistics for primary fetal samples. Log2FC is calculated as Growth plate vs. epiphyseal (GP-vs-EP). Columns O-AG: Normalized transcript counts for the indicated gene in the indicated sample. (c) DEG gene-set enrichments. The (up to) top 30 significantly-enriched gene sets obtained using sets of genes up-regulated in the given tissue. (d) Testing shared direction

of differential expression between primary and in vitro hESC-derived samples. (e) Differential expression information for differentially expressed in at least one of four cell types. Column 'A' denotes the classification of different TFs. (f) Differential expression analysis run only on 'Batch 2' samples, for which paired ATAC-seq libraries were generated. (g) Sets of that were DEG in the Batch 2 analysis.

• Supplementary file 2. Sequencing information for the in vitro chondrogenic time course RNA-seq run. (a) DEG data for hESC-derived (in vitro) chondrocyte samples after 4 weeks of in vitro differentiation. Columns C-H: DESeq2 statistics. Log2FC is calculated as BMP-vs-TGF. Columns I-N: Normalized transcript counts for the indicated gene in the indicated sample. (b) DEG data for hESC-derived (in vitro) chondrocyte samples after 8 weeks of in vitro differentiation. Columns C-H: DESeq2 statistics. Log2FC is calculated as BMP-vs-TGF. Columns I-N: Normalized transcript counts for the indicated gene in the indicated sample. (c) DEG data for hESC-derived (in vitro) chondrocyte samples after 12 weeks of in vitro differentiation. Columns C-H: DESeq2 statistics. Log2FC is calculated as BMP-vs-TGF. Columns I-N: Normalized transcript counts for the indicated gene in the indicated sample. (d) DEG gene-set enrichments in each lineage and time. The (up to) top 30 significantly-enriched gene sets were obtained using sets of genes up-regulated in the indicated tissue and time (e) Differential expression information for differentially expressed at 4 weeks. (f) Differential expression information for differentially expressed at 8 weeks. (g) Differential expression information for differentially expressed at 12 weeks. (h) Allocation of in the TGF lineage which were differentially expressed in one or more timepoints during in vitro differentiation. (i) Allocation of in the BMP lineage which were differentially expressed in one or more timepoint during in vitro differentiation.

• Supplementary file 3. Sequencing information for ATAC-seq runs. (a) Quality summary statistics for ATAC-seq samples. (b) Sets of differentially-accessible peaks, separated by their lineage bias. hg19 coordinates shown. (c) GREAT results for differentially-accessible peak sets. Description of column names and tests are available from the GREAT website. (d) HOMER de-novo motif analysis results for DA peaks in growth plate chondrocytes. (e) HOMER de-novo motif analysis results for DA peaks in articular chondrocytes. (f) Results of motif testing of all differentially-expressed transcription factors with the indicated DA peaksets.

• Supplementary file 4. Gene regulatory behaviors. (a) Summarized statistics comparing genes grouped by regulatory behavior. (b) Significant genes assigned to cluster 1 - 'Poorly explained.' DESeq2 DEG statistics for all genes are shown, along with the amount of variance in gene expression which was ascribed to either enhancer cis-regulatory scores, promoter accessibility, the interaction term, and variance which could not be explained, 'Unexplained.' (c) Significant genes assigned to cluster 2 - 'Combo-centric.' (d) Significant genes assigned to cluster 3 - 'Promoter-centric.' (e) Significant genes assigned to cluster 4 - 'Enhancer-centric.' (f) The set of differentially-expressed transcription factors for which motif enrichments were tested in regulatory sequences associated with genes falling in different regulatory groups. Columns I-K indicates whether a significant enrichment was observed for promoter-centric, enhancer-centric, or combo-centric genes. Column L indicates the direction of the differential expression of a given TF. (g) Results of motif enrichment testing for promoters of promoter-centric genes. Columns E-G refers to the enrichment statistics for BMP-biased DEGs. Columns I-K refers to enrichment statistics for TGF-biased DEGs. Sets of TFs are grouped based on their lineage-specificity behaviors. (h) Results of motif enrichment testing for putative enhancers of enhancer-centric genes. Columns E-I refers to the enrichment statistics for BMP-biased DEGs. Columns J-O refers to enrichment statistics for TGF-biased DEGs. 'GENOME_TAG' refers to enrichment results when using the entire genome as a background set. Sets of TFs are grouped based on their lineage-specificity behaviors. (i) Results of motif enrichment testing for putative enhancers of combo-centric genes. Columns E-I refers to the enrichment statistics for BMP-biased DEGs. Columns J-O refers to enrichment statistics for TGF-biased DEGs. 'GENOME_TAG' refers to enrichment results when using the entire genome as a background set. Sets of TFs are grouped based on their lineage-specificity behaviors.

• Supplementary file 5. Overlapping ChIP-seq datasets with DA peaksets. (a) Overlapping ChIP-seq datasets with DA peaksets. Accessions listed in this sheet for RELA and RUNX2 correspond to those accessions listed in *Supplementary file 5b- e*. (b) Overlaps of RELA ChIP-seq datasets with BMP-biased peak sets. Column A: Whether or not the indicated region (Columns B-D) contained a FIMO-predicted motif hit for RELA. 'Num_hits': the number of ChIP-seq peaks (pooled across individual datasets) which overlapped the given region. 'Hits':' the individual overlapping ChIP-seq peaks, separated by '@'. 'sources' and 'ID' refers to the individual datasets corresponding to each ChIP-seq peak separated by '@' in 'Hits' column. (c) Overlaps of RELA ChIP-seq datasets with TGF-

biased peak sets. (d) Overlaps of RUNX2 ChIP-seq datasets with BMP-biased peak sets. (e) Overlaps of RUNX2 ChIP-seq datasets with TGF-biased peak sets.

• Supplementary file 6. Oligonucleotides used in this report.

• MDAR checklist

## Data availability

The raw ATAC-seq and RNA-seq datasets reported in this paper are available on GEO under accession GSE195688.

The following dataset was generated:

| Author(s) | Year | Dataset title | Dataset URL | Database and Identifier |
|---|---|---|---|---|
| Richard D, Pregizer S, Venkatasubramanian D, Muthuirulan P, Liu Z, Capellini TD, Craft AM | 2022 | Lineage-Specific Differences and Inference of Regulatory Networks Governing Human Chondrocyte Development | http://www.ncbi.nlm.nih.gov/geo/query/acc.cgi?acc=GSE195688 | NCBI Gene Expression Omnibus, GSE195688 |

The following previously published dataset was used:

| Author(s) | Year | Dataset title | Dataset URL | Database and Identifier |
|---|---|---|---|---|
| Richard Liu, Willen C | 2020 | Regulatory constraint and selection during human knee evolution drive modern osteoarthritis risk | https://www.ncbi.nlm.nih.gov/geo/query/acc.cgi?acc=GSE122877 | NCBI Gene Expression Omnibus, GSE122877 |

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
