## [Editor Report]

In this study the authors mapped chromatin accessibility in hESC derived chondrocyte lineages and mouse embryonic chondrocytes using ATAC-sequencing and revealed lineage-specific gene regulatory networks. They further validated the functional interactions of two transcription factors, Runx2 and RELA, with their predicted genomic targets. The significance of study is to help us understand chondrocyte differentiation mechanism.

---

## [Decision Letter]

**Decision letter after peer review:**

Thank you for submitting your article "Lineage-Specific Differences and Regulatory Networks Governing Human Chondrocyte Development" for consideration by *eLife*. Your article has been reviewed by 2 peer reviewers, and the evaluation has been overseen by a Reviewing Editor and Mone Zaidi as the Senior Editor. The reviewers have opted to remain anonymous.

Based on the discussion by two reviewers, it seems that works presented in this manuscript in general is interesting.

1) The authors need to add more time points, especially the early time point, for their studies.

2) Using single cell RNA sequencing technique will add additional value, but it would be interesting to publish current works (with more time points), even without single cell RNA sequencing approach. The authors may discuss this point as prospective studies.

In addition, the authors should consider addressing several other concerns raised by the reviewers.

*Reviewer #2 (Recommendations for the authors):*

In the current study, the authors aim to apply their in vitro differentiation approach to address the large gap in understanding articular and cartilage development mechanisms in humans. Based on their previous study, hESC were cultured under specified conditions and directed into terminal differentiated cartilage tissue with a similar organization as fetal cartilage. Bulk RNA sequencing and ATAC-sequencing analysis were performed for hESC-derived cartilage tissue and fetal cartilage. Comprehensive bioinformatic analyses were performed to identify new lineage-specific genes and new potential regulation networks. New lineage-specific genes were verified using RNA Scope or immunohistochemical staining. A putative new target of transcription regulation for cartilage development was validated using Chip-qPCR. The study has the following issues.

1. Although the study is aiming to provide an in vitro model for studying human cartilage development. Only samples of one time point were analyzed for both hESC-derived cartilage and human fetal cartilage, including terminal differentiated hESC-derived cartilage and E67 fetal donor femur cartilage. The information can hardly be defined as a developmental biology analysis.

2. Single cells RNA sequencing is becoming the current standard for most sequencing-related studies. The bulk RNA sequencing analysis used in this study provided much less information and is below the current standard. The sequencing results are, therefore, less valuable as a resource for other researchers in the field.

3. The validation evidence provided in the study is too weak. Transgenic mouse models are strongly recommended to validate the transcript regulation target identified in the analysis.

---

## [Author Response]

Reviewer #2 (Recommendations for the authors):1. Although the study is aiming to provide an in vitro model for studying human cartilage development. Only samples of one time point were analyzed for both hESC-derived cartilage and human fetal cartilage, including terminal differentiated hESC-derived cartilage and E67 fetal donor femur cartilage. The information can hardly be defined as a developmental biology analysis.

We thank this Reviewer for this terrific suggestion and we have now included a new set of experiments in which we performed bulk RNA sequencing analysis on early, mid, and late stage cartilage tissues during differentiation. The late stage cartilage tissues showed similar gene expression differences to those we identified in the initial transcriptomic dataset (Results, Page 6). We present the biological processes associated with DEGs identified at early and mid-stages of differentiation, with an additional focus on the transcription factors both unique to each developmental stage, and those that are shared at all stages during articular or growth plate cartilage differentiation in vitro. These new data can be found in Figure 1—figure supplement 3 and supplementary file 2.

2. Single cells RNA sequencing is becoming the current standard for most sequencing-related studies. The bulk RNA sequencing analysis used in this study provided much less information and is below the current standard. The sequencing results are, therefore, less valuable as a resource for other researchers in the field.

We very much appreciate this suggestion, as we also believe that single cell transcriptomic data, particularly of human cartilage, is highly valuable as a resource for other researchers. This work was initiated by a senior fellow in the Craft laboratory. However, we must respectfully disagree on the bulk RNA sequencing providing less information, particularly for our goal of defining differential transcription factor expression. Single cell RNA sequencing is less powered to detect statistically significant differences in low copy number transcripts, notably including transcription factors, due to significantly lower read depth than that which can be obtained by bulk RNA sequencing. Methods for integrating bulk RNA-seq and ATAC-seq were available when the present work was performed, while methodology to obtain matched RNA- and ATAC-seq from individual cells, especially those that must be isolated from dense extra-cellular matrix environments, is currently being optimized. We think the bulk data are interesting, important, and appropriate to publish now. However, we agree with this Reviewer about the importance of adding single cell data when we are confident those data are as robust and reproducible as our bulk data.

3. The validation evidence provided in the study is too weak. Transgenic mouse models are strongly recommended to validate the transcript regulation target identified in the analysis.

We agree that the function of many of our newly identified hits should be validated in vivo using transgenic animal models, and we highlight this limitation in the Discussion (Page 13). We have referred to published studies for several of the transcription factors we found in our analyses. Our discovery of known chondrogenic factors in our analyses suggests that the new targets we identified may also have biological functions in chondrocyte biology. We strongly believe that the resource we are providing to the community will lead to such investigations.